# B3galt5 functions as a PXR target gene and regulates obesity and insulin resistance by maintaining intestinal integrity

Jinhang Zhang[1,7], Ya Huang[1,2,7], Hong Li[1], Pengfei Xu[3], Qinhui Liu[1], Yang Sun[4], Zijing Zhang[1], Tong Wu[1], Qin Tang[1], Qingyi Jia[1], Yan Xia[1], Ying Xu[1], Xiandan Jing[1], Jiahui Li[1], Li Mo[5], Wen Xie[3], Aijuan Qu [6], Jinhan He [1] ✉ & Yanping Li [1] ✉

Pregnane X receptor (PXR) has been reported to regulate glycolipid metabolism. The dysfunction of intestinal barrier contributes to metabolic disorders. However, the role of intestinal PXR in metabolic diseases remains largely unknown. Here, we show that activation of PXR by tributyl citrate (TBC), an intestinal-selective PXR agonist, improves high fat diet (HFD)-induced obesity. The metabolic benefit of intestinal PXR activation is associated with upregulation of β-1,3 galactosyltransferase 5 (B3galt5). Our results reveal that B3galt5 mainly expresses in the intestine and is a direct PXR transcriptional target. B3galt5 knockout exacerbates HFD-induced obesity, insulin resistance and inflammation. Mechanistically, B3galt5 is essential to maintain the integrity of intestinal mucus barrier. B3galt5 ablation impairs the *O*-glycosylation of mucin2, destabilizes the mucus layer, and increases intestinal permeability. Furthermore, B3galt5 deficiency abolishes the beneficial effect of intestinal PXR activation on metabolic disorders. Our results suggest the intestinal-selective PXR activation regulates B3galt5 expression and maintains metabolic homeostasis, making it a potential therapeutic strategy in obesity.

The prevalence of obesity is increasing worldwide, contributing substantially to the global burden of other related chronic metabolic diseases, such as type 2 diabetes, non-alcoholic fatty liver diseases and cardiovascular diseases[1]. Obesity is often accompanied by several alterations of hormonal, inflammatory, and disordered lipid and glucose levels[2]. Accumulating evidence has revealed the occurrence of obesity and related metabolic syndrome focusing on complex interactions between numerous heritable genes and environmental factors[3]. However, one of the poorly understood features of the metabolic syndromes is their association with intestinal barrier

dysfunction, which leads to increased intestinal permeability and the transference of microbial molecules into the portal vein blood and liver[4,5]. This influx of immune-stimulatory microbial components or metabolites into the circulation contributes to an enhanced risk of obesity and related metabolic disorders[4].

The intestinal barrier is a selective physical barrier that facilitates the absorption of fluids and nutrients while preventing the translocation of potentially harmful luminal antigens into the circulation[6]. Among the intestinal barrier, the integrity of the mucus barrier forms the first line of defense against bacteria invasion and is essential for health

[1]Department of Pharmacy, Institute of Metabolic Diseases and Pharmacotherapy, National Clinical Research Center for Geriatrics, West China Hospital, Sichuan University, Chengdu, Sichuan Province, China. [2]Department of Pharmacy, GuiQian International General Hospital, Guiyang, China. [3]Center for Pharmacogenetics and Department of Pharmaceutical Sciences, University of Pittsburgh, Pittsburgh, PA, USA. [4]Department of Gastroenterology, The First Affiliated Hospital of Kunming Medical University, Yunnan Institute of Digestive Disease, Kunming, Yunnan Province, China. [5]Center of Gerontology and Geriatrics, West China Hospital of Sichuan University, Chengdu, China. [6]Department of Physiology and Pathophysiology, School of Basic Medical Sciences, Capital Medical University, Beijing, P.R. China. [7]These authors contributed equally: Jinhang Zhang, Ya Huang. ✉e-mail: jinhanhe@scu.edu.cn; liyanping_512@163.com

maintenance[7,8]. Mucus is a complex gel composed of mucin glycoproteins, water, bioactive peptides, and microbial metabolites. The major glycoprotein of mucus is mucin 2 (Muc2), which forms homodimers in the endoplasmic reticulum and is glycosylated in the Golgi apparatus. Glycosylation is a key step in the production of functional mature mucins, and O-linked glycans account for up to 80% of the molecular weight of mucins[9]. A high number of glycosyltransferases, such as β1,3-N-acetylglucosaminyltransferase 5, 2-alphaL-fucosyltransferase 2, and β−1,3 galactosyltransferase 5 (B3galt5), is responsible for the generation of various mucins O-glycosylation, which is vital for the protective and bacterial binding capacity of mucus[10,11]. Dysfunction of mucus barrier may contribute to the development of infection, autoimmune, and metabolic disorders[12]. Previous study has shown that HFD-induced obese mice developed dysbiosis and affected Muc2 expression and secretion[13–15]. High fructose intake induced a loss of mucus thickness in the colon and impaired gut permeability[16]. High saturated fatty acids diets present risk factors for obesity and associated metabolic disorders by affecting Muc2 production[17]. These studies suggest the importance of Muc2 in obesity and metabolic diseases. However, whether and how glycosylation of Muc2 directly protects against obesity and metabolic syndrome remain to be investigated.

Pregnane X receptor (PXR) is originally identified as a xenobiotic receptor to regulate drug metabolism as it is highly expressed in the liver and intestine[18,19]. Interestingly, it has been demonstrated that PXR plays crucial roles in glucose, lipid, and bile acid metabolism, making it a potential therapeutic target in obesity and type 2 diabetes[20,21]. However, PXR appeared to have conflicting functions in lipid metabolism. Whole-body PXR deficiency improved HFD-induced obesity, and activation of PXR exacerbated hypertriglyceridemia and insulin resistance[22,23]. In contrast, treating HFD-induced mice with whole-body PXR agonist pregnenolone-16a carbonitrile (PCN) mitigated obesity and insulin resistance[24]. Besides, selective activation of PXR in the liver leads to lipid accumulation, and contributes to hepatic steatosis and insulin resistance[25]. These conflicting results suggest that PXR may have multiple regulatory mechanisms or different functions in different tissues. Although PXR is highly expressed in intestine, the function of intestinal PXR on metabolic diseases remains largely unknown.

Here, we provide the first evidence that intestinal-selective activation of PXR by TBC could alleviate obesity and insulin resistance of HFD-fed mice. We identified TBC could significantly upregulate intestinal B3galt5 expression using RNA sequencing, and further confirmed that B3galt5 as a direct transcriptional target gene of PXR. Mechanistically, we found that B3galt5 mediated mucus O-glycosylation and rendered it resistant to proteolytic degradation, thus supporting the integrity and function of the intestinal barrier. Whole-body and intestinal-specific knockout of B3galt5 both aggravated HFD-induced obesity, insulin resistance, and inflammation. B3galt5 knockout also abolished the protective effects of TBC on obesity and insulin resistance. Our results demonstrate the importance of the PXR-B3galt5 axis in metabolic homeostasis maintenance, making it a potential therapeutic strategy in obesity.

## Results

### Activation of intestinal PXR ameliorates HFD-induced obesity and insulin resistance
To explore the potential function of intestinal PXR on metabolic diseases, we first tested whether tributyl citrate (TBC) was an intestinal selective PXR agonist as previously reported[26]. As shown in Fig. 1a, b, TBC only increased the expression of Cyp3a11, a well-known PXR target gene, in the intestine. In contrast, pregnenolone 16α-carbonitrile (PCN), a whole-body PXR agonist, increased Cyp3a11 expression in both the liver and intestine (Fig. 1a, b). The increase of Cyp3a11 by TBC appeared to be PXR dependent because the ablation of PXR completely abolished its effect (Fig. 1c, d). These results confirmed that TBC is indeed a selective agonist for intestinal PXR.

To evaluate the role of intestinal PXR on metabolic disorders, we fed WT mice with TBC supplemented in HFD at 0.05% (w/w) for 12 weeks. Upon HFD feeding, TBC treatment led to significantly lower weight gain (Fig. 1e) and decreased ratio of fat pad weight to body weight (Fig. 1f-h), as well as smaller numbers of large adipocytes in epididymal white adipose tissue (eWAT), inguinal white adipose tissue (iWAT) and brown adipose tissue (BAT) (Fig. 1i, j). TBC treatment also improved glucose tolerance and insulin tolerance (Fig. 1k, l). The improved insulin sensitivity seen in TBC-treated mice was associated with significantly lower serum levels of insulin and leptin (Fig. 1m, n). The effect of TBC is PXR dependent, because ablation of PXR totally abolished the TBC-decreased body weight and improved insulin sensitivity (Figure S1a-i). These results suggest the selective activation of intestinal PXR can alleviate obesity and insulin resistance in HFD-fed mice.

### PXR upregulates the expression of B3galt5 as a target gene
To understand the mechanism of intestinal PXR activation improved metabolic disorder, we used RNA sequencing to screen PXR-regulated genes (Fig. 2a). Activation of PXR upregulated several genes, including known PXR target genes such as Cyp2c55 and Cyp3a11. We found that B3galt5 was the second highest upregulated gene by PXR activation (Fig. 2a). The volcano plot also showed B3galt5 was significantly elevated in mice colon (Figure S2a). To further confirm the result of RNA sequencing, we measure the expression of B3galt5 by qPCR. As shown in Fig. 2b, c, PXR activation by either PCN or TBC strongly upregulated mRNA expression of B3galt5 in WT mice. However, the effect of PCN or TBC on B3galt5 mRNA expression was completely abolished in PXR knockout mice (Fig. 2c). Consistently, the protein expression of B3galt5 was upregulated by either PCN or TBC in WT mice, but not in PXR knockout mice (Fig. 2d and Figure S2b-d). Constitutive androstane receptor (CAR) is a sister receptor of PXR, and shares reciprocal regulated genes[27]. However, activation of CAR did not affect the expression of B3galt5 (Figure S2e). Activation of PXR had little effect on the expression of other galactosyltransferases (Figure S2f-i).

To determine whether B3galt5 is a direct transcriptional target of PXR, we inspected the mouse B3galt5 promoter and uncovered a DR4-type (direct repeat spaced by four nucleotides) of nuclear receptor-response element (gccAGGTCAggaaAGATCAgcc) (Fig. 2e). Electrophoretic mobility shift assay (EMSA) showed that the PXR/retinoic X receptor (RXR) heterodimer could bind to this DR4 site, parallel to the positive control Cyp3a23, a known PXR target (Fig. 2e). The binding of B3galt5/DR4 to PXR was specific, as evidenced by competitively inhibition of binding by excess B3galt5/DR4, but not by the mutant B3galt5/DR4 (Fig. 2e). Further luciferase reporter assays revealed PXR was able to induce a 2.4-fold increase in B3galt5 promoter activity upon PCN activation, which was completely abolished by mutated B3galt5 DR4 promoter (Fig. 2f). These in vitro results were subsequently confirmed by chromatin immunoprecipitation (ChIP) assays in intestinal lysate of TBC-treated mice to evaluate the binding of PXR binding to the B3galt5 promoter in vivo. As shown in Fig. 2g, TBC treatment promoted the recruitment of PXR onto the B3galt5 promoter. In addition, B3galt5 mRNA and protein expression were significantly upregulated after PXR activation by either TBC or rifampicin in LS174T cells, a human colon adenocarcinoma cell-line (Fig. 2h-j). Together, these results indicate that PXR upregulated the expression of B3galt5 as a target gene in both mice and human beings.

### B3galt5 is specifically expressed in colon and downregulated during obesity
To establish the function of B3galt5, we characterized the tissue distribution of B3galt5 by qPCR and western blot analysis. The results showed that B3galt5 is highly expressed in colon and barely expressed in other tissues, such as liver and adipose tissues (Fig. 3a, b). We subsequently tested the specific cell types in colon expressing B3galt5

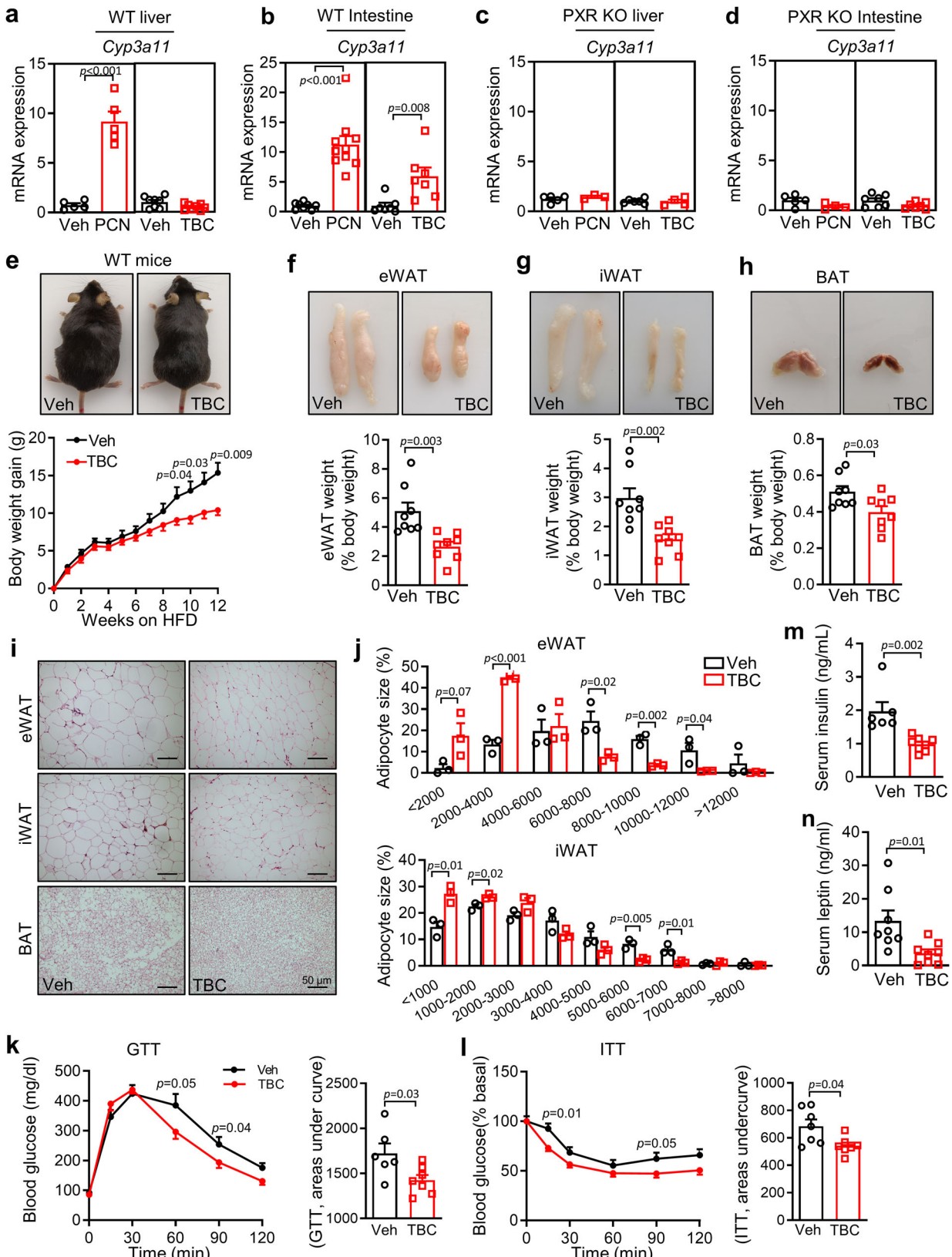

using immunofluorescence staining. As shown in Fig. 3c, B3galt5 was mostly expressed in colonic epithelial cells, especially goblet cells (Fig. 3c), as evidenced by the highly overlap of B3galt5 with the epithelial cell marker β-catenin and the goblet cell marker *Ulex europaeus* agglutinin 1 (UEA-1), respectively.

To investigate whether B3galt5 is involved in the pathogenesis of metabolic diseases, we first measured its expression in the intestine of mice with metabolic disorders. Both the mRNA and protein levels of intestinal B3galt5 were observed remarkably decreased in HFD-induced obesity mice and *ob/ob* mice (Fig. 3d-g), as well as high fructose and high cholestrol-induced mouse model (Figure S3a, b).

**Fig. 1 | TBC, an intestinal selective PXR agonist, ameliorates HFD-induced obesity and insulin resistance. a, b** The mRNA expression of Cyp3a11, a typical target gene of PXR in the liver (**a**; $n = 5$ for Veh and PCN, $n = 8$ for Veh and TBC) and intestine (**b**; $n = 8$ for Veh, $n = 10$ for PCN, $n = 7$ for Veh and TBC) of WT mice treated with PCN (40 mg/kg) once every 8 hours for three times. **c, d** The mRNA expression of Cyp3a11 in the liver (**c**; $n = 5$ for Veh, $n = 3$ for PCN, $n = 6$ for Veh, $n = 4$ for TBC) and intestine (**d**; $n = 5$ for Veh, $n = 4$ for PCN, $n = 7$ for Veh and TBC) of PXR knockout mice treated with PCN or TBC. **e** Appearance (top) and growth curve (bottom) of vehicle (Veh) and TBC-treated mice. WT mice fed with high-fat diet (HFD) supplemented with vehicle or 0.05% TBC for 12 weeks ($n = 8$ per group). **f–h** Representative photographs and the ratio of fat depots to body weight of eWAT (**f**), iWAT (**g**), and BAT (**h**) ($n = 8$ per group). **i** H&E staining of adipose tissues. Scale bar: 50 μm. **j** Distribution of adipocyte size of eWAT and iWAT ($n = 3$ per group). **k, l** Blood glucose concentrations during GTT (2 g/kg; **k**) and ITT (0.75 U/kg; **l**) in vehicle and TBC-treated WT mice ($n = 6$ for Veh, $n = 7$ for TBC). **m** Serum insulin levels of vehicle and TBC treated mice ($n = 6$ for Veh, $n = 8$ for TBC). **n** Serum leptin levels vehicle and TBC treated mice ($n = 8$ per group). PXR KO PXR whole-body knockout mice, WT wild type, eWAT epididymal white adipose tissue, iWAT inguinal white adipose tissue, BAT brown adipose tissue, GTT glucose tolerance test, ITT insulin tolerance test. Data are mean ± SEM. All the data were assessed form normal distribution before statistical analysis. Significance was analyzed using two-way analysis of variance (ANOVA) with Sidak's multiple comparisons test (Fig. 1e, k, and l). The remaining statistical differences were determined using unpaired two-tailed Student's t test. Source data are provided as a Source Data file.

Consistently, B3galt5 expression was downregulated in intestine biopsies from obese patients comparing to normal controls (Fig. 3h, i). Moreover, the colonic B3galt5 levels were negatively correlated with BMI (Fig. 3j). Of note, other intestinal O-glycosyltransferase expression was little affected at obese status (Figure S3c-e). Thus, our data suggest that B3galt5 is compromised in both obese mouse models and patients, possibly playing an important role in metabolic diseases as a downstream regulator of PXR.

### B3galt5 whole-body deficiency exacerbates HFD-induced obesity, insulin resistance and inflammation in mice

Since B3galt5 shows a potential role in metabolic diseases, we generated B3galt5 whole-body knockout (*B3galt5*$^{-/-}$) mice by CRISPR-Cas9-mediated gene targeting (Figure S4a-b). The expression of B3galt5 was substantially reduced in the intestine of *B3galt5*$^{-/-}$ mice as compared with WT mice. On 12 weeks of chow diet (CD), WT and *B3galt5*$^{-/-}$ mice showed similar body weight and fat weight (Figure S4c-d). However, HFD for a duration of only 4 weeks had already displayed significantly more body weight in *B3galt5*$^{-/-}$ mice than WT mice (Fig. 4a). After 12 weeks of HFD feeding, *B3galt5*$^{-/-}$ mice were visibly bigger and exhibited an increase in eWAT, iWAT and BAT than WT mice (Fig. 4a-d). H&E staining and its quantitative evaluation showed increased adipocyte morphology features and greater amounts of large adipocytes in *B3galt5*$^{-/-}$ mice comparing to the controls (Fig. 4e; Figure S4e).

Because obesity is often accompanied with insulin resistance and glucose intolerance[28], we performed a series of metabolic tests to examine the effect of the whole-body loss of *B3galt5* on glucose homeostasis. On a chow diet, WT and *B3galt5*$^{-/-}$ mice showed comparable glucose and insulin tolerance (Figure S4f). When challenged with HFD, *B3galt5*$^{-/-}$ mice showed impaired glucose tolerance and insulin resistance as compared to WT mice (Fig. 4f and Figure S5a). Aggravated insulin resistance was further confirmed by remarkable suppression of Akt phosphorylation in the liver, eWAT and skeletal muscle in *B3galt5*$^{-/-}$ mice after insulin administration (Fig. 4g; Figure S5b-d). The circulating levels of insulin and leptin were significantly elevated in *B3galt5*$^{-/-}$ mice, further verifying the impaired insulin sensitivity (Fig. 4h).

Obesity is often accompanied with impaired energy metabolism[29]. To determine whether *B3galt5*$^{-/-}$ affects whole-body fuel metabolism and substrate preference, we used metabolic cages to measure food, $O_2$, and $CO_2$ consumption. Compared to WT, *B3galt5*$^{-/-}$ mice showed significantly decreased oxygen consumption, carbon dioxide production, and energy expenditure, despite no difference in food intake was observed (Fig. 4i, j; S5e, f). These results indicate that whole-body B3galt5 deficiency leads to impaired energy production. Respiration exchange rate (RER) was comparable between *B3galt5*$^{-/-}$ and WT mice, suggesting similar substrate utilization in both groups (Figure S5g). We noticed significantly inhibited lipolysis in adipose depots after B3galt5 knockout, possibly due to the impaired intrinsic effect, such as thermogenesis, of adipose tissue (Figure S5h-l).

In obesity, excessive adipose expansion causes adipose dysfunction and inflammation, which subsequently results in increased inflammation levels in other tissues, such as liver[30]. Therefore, we analyzed the mRNA and protein levels of proinflammatory cytokines in liver and adipose tissues from HFD-fed *B3galt5*$^{-/-}$ mice. *B3galt5*$^{-/-}$ mice showed significant increases in the pro-inflammatory factors, including Il-1β, Il-6, Mcp-1 and Tnf-α, in the liver and adipose tissue compared to WT mice (Fig. 5a-d). Consistently, we found *B3galt5*$^{-/-}$ mice displayed increased number of F4/80-positive macrophages in both liver and eWAT using immunohistochemistry (Fig. 5e, f). The above results have indicated increased infiltration of adipose tissue macrophages (ATMs) in *B3galt5*$^{-/-}$ mice. ATMs are usually divided into classically activated macrophages (M1), which secret proinflammatory cytokines, and alternatively activated macrophages (M2), which secret anti-inflammatory cytokines[31]. Using flow cytometry, we found significantly increased percentage of Cd11b$^+$/F4/80$^+$ double-positive macrophages and greatly reduced percentage of Cd206$^+$/Cd11b$^+$/F4/80$^+$ triple-positive M2 macrophages of HFD-fed *B3galt5*$^{-/-}$ mice (Fig. 5g; Figure S6a-b). As a result, the ratio of M1 to M2 macrophages was notably increased in eWAT of *B3galt5*$^{-/-}$ mice comparing to the WT (Fig. 5g). Furthermore, *B3galt5*$^{-/-}$ mice displayed more severe liver steatosis as shown by the elevated serum ALT/AST level and hepatic lipid deposition (Figure S7a-b). Together, these findings demonstrate that whole-body B3galt5 deficiency sensitizes mice to HFD-induced obesity, insulin resistance and inflammation.

### B3galt5 deficiency increases the permeability of the intestinal barrier by disrupting mucus O-glycosylation

Studies have revealed that the compromised permeability of the intestinal barrier may be responsible to metabolic diseases[6]. As previous studies, we found that HFD feeding could reduce the intestinal mucus layer and increase intestinal permeability (Figure S7c-d). To understand the possible mechanism of B3galt5 deficiency-aggravated obesity, we examined the changes of intestinal permeability by FITC-dextran gavage in WT and *B3galt5*$^{-/-}$ mice. On a CD diet, *B3galt5*$^{-/-}$ mice showed mildly elevated FITC-dextran concentration in serum compared with WT mice. However, upon HFD challenge, *B3galt5*$^{-/-}$ mice showed markedly increased serum concentration of FITC-dextran as compared with WT mice (Fig. 6a). The mucus layer represents the first defense line for intestinal barrier and closely relates to metabolic diseases[32]. Alcian blue staining of colonic sections showed a clear and strong mucus layer in WT mice; in contrast, the thickness of the inner mucus layer was significantly reduced in *B3galt5*$^{-/-}$ mice (Fig. 6b). Consistently, immunofluorescence staining of Muc2 revealed a severely impaired colonic mucus layer in *B3galt5*$^{-/-}$ mice (Fig. 6b). To examine whether the reduction of the mucus layer in *B3galt5*$^{-/-}$ mice was due to the disruption of mucin, we scraped and extracted colonic luminal mucus, and analyzed Muc2 expression by a composite agarose-polyacrylamide gel electrophoresis (AgPAGE). As shown in Fig. 6c, Muc2 band was found significantly decreased in *B3galt5*$^{-/-}$ mice than in WT mice. Proteases contribute to the degradation to mucus layer[33]. To determine the ability of mucin resisting to protease degradation, we treated mucin extracted from WT and *B3galt5*$^{-/-}$ mice with

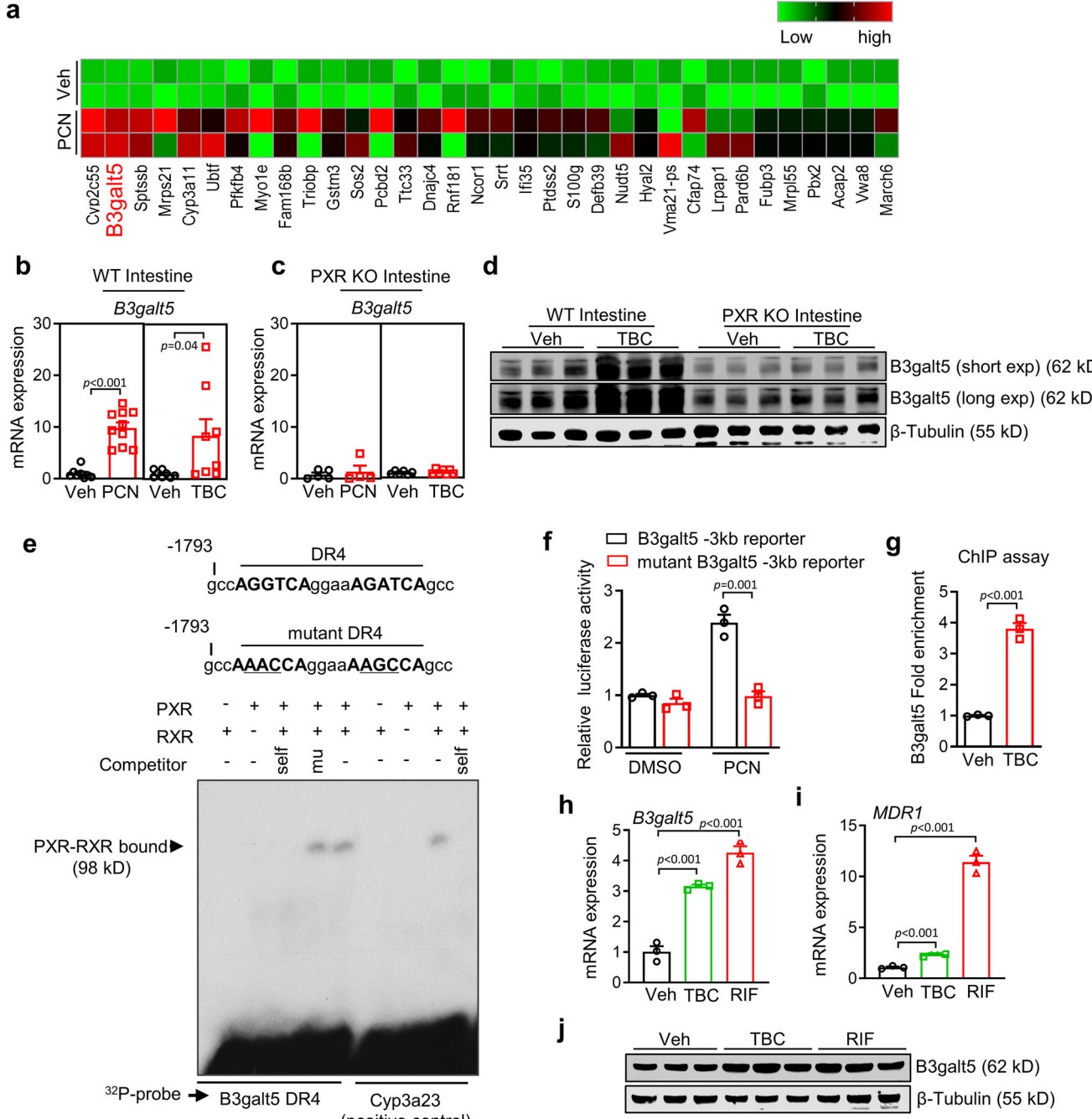

**Fig. 2 | B3galt5 is a direct transcriptional target gene of PXR. a** Heat map representation of genes deferentially expressed in the intestine of 8-week-old mice treated with vehicle (Veh) and PCN. Mice were treated with PCN (40 mg/kg) once every 8 hours for three times ($n = 2$ per group). **b** Intestinal mRNA expression of B3galt5 in WT mice treated with PXR agonists PCN and TBC ($n = 8$ for Veh, $n = 10$ for PCN, $n = 8$ for Veh, $n = 8$ for TBC). Significance was determined using unpaired two-tailed Student's t test. **c** Intestinal mRNA expression of B3galt5 in PXR knockout mice treated with PCN and TBC ($n = 5$ for Veh, $n = 4$ for PCN, $n = 6$ for Veh, $n = 7$ for TBC). **d** Intestinal protein level of B3galt5 in WT and PXR knockout mice treated with TBC, β-Tubulin was used as the loading control. **e** Electrophoretic mobility shift assay (EMSA) showing that PXR binds to the −1793 to −1814 region of B3galt5 promoter. **f** Luciferase assay of HEK293 cells co-transfected with B3galt5 or mutant

B3galt5 and treated with PCN for 24 h. The luciferase activity was normalized to β-gal ($n = 3$ per group). **g** ChIP assay of recruitment of PXR to the B3galt5 promoter ($n = 3$ per group). **h, i** The mRNA expression of B3galt5 (**h**) and MDR1 (**i**) in LS174T cells treated with TBC (5 µM) or rifampicin (RIF, 20 µM) for 48 h ($n = 3$ per group). **j** The protein levels of B3galt5 in LS174T cells treated with TBC or RIF, β-Tubulin was used as the loading control. Data are mean ± SEM. PXR KO PXR whole-body knockout mice, WT wild type, TBC tributyl citrate, ChIP chromatin immunoprecipitation, RIF rifampicin. The data sets (Fig. 1f-i) were analyzed using non-parametric approaches and the statistical differences between groups were determined using one-way ANOVA with post-hoc Tukey test. Source data are provided as a Source Data file.

different concentration of pronase. AgPAGE analysis followed by PAS staining showed that pronase had little effect on WT mucus (Fig. 6d). In contrast, pronase treatment resulted in a dramatic decrease in high MW PAS-stained bands compared with non-treated mucus in *B3galt5⁻/⁻*

mice (Fig. 6d). We further used another two protease, O-glycoprotease (OgpA) from *A. muciniphila* and the secreted protease of C1 esterase inhibitor (StcE) and found mucin from *B3galt5⁻/⁻* mice was more sensitive to degradation (Fig. 6e)[34,35]. These results suggest that B3galt5

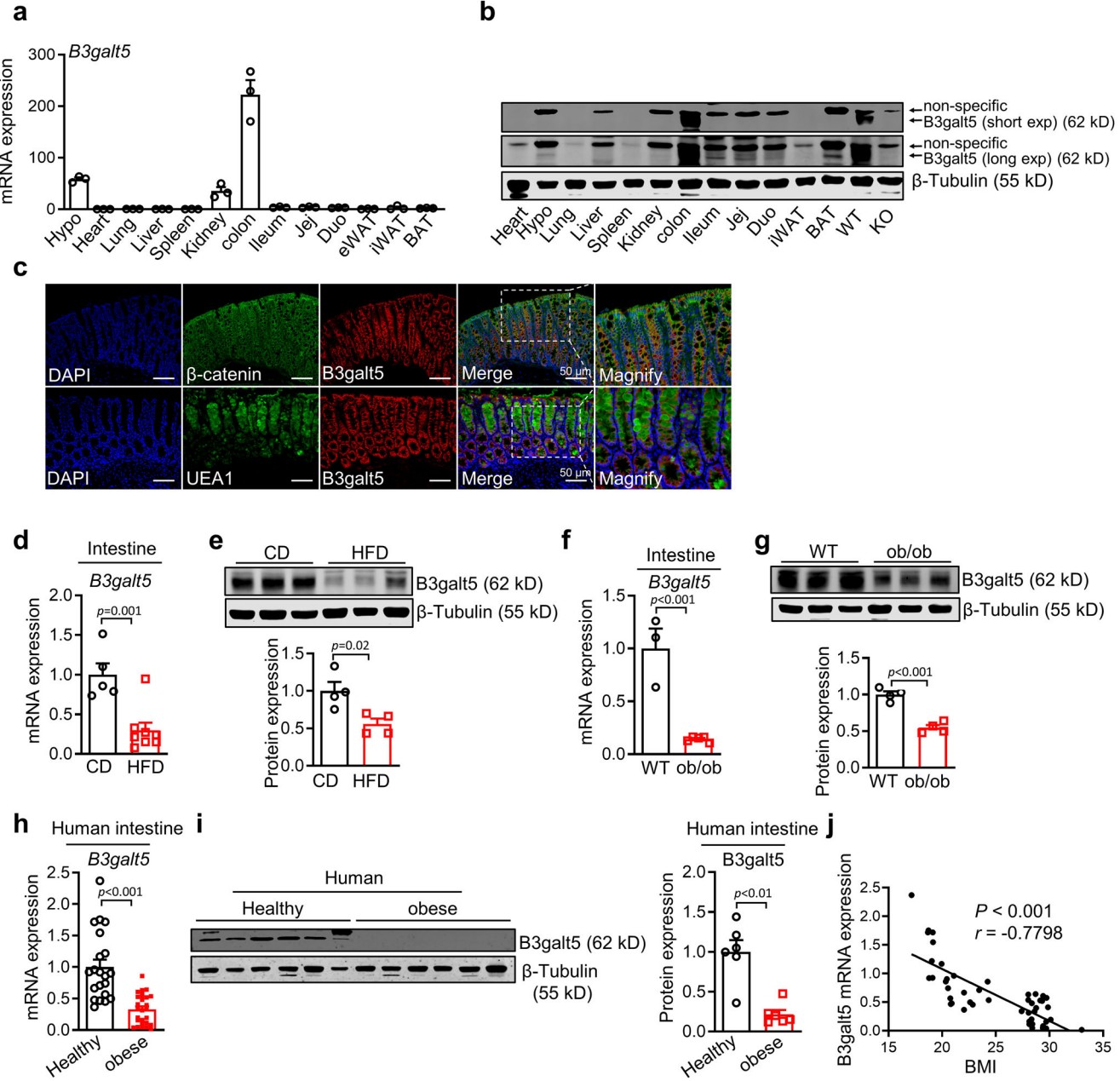

**Fig. 3 | B3galt5 is specifically expressed in colon and downregulated during obesity. a** The mRNA expression of B3galt5 in different tissues of 8-week-old male mice ($n = 3$ per group). **b** The protein levels of B3galt5 in different tissues of 8-week-old male mice. **c** Co-immunofluorescent staining of B3galt5 with intestinal epithelial cells marker β-catenin (top) or goblet cells marker UEA1 (bottom) in mice colon. Scale bar: 50 μm. **d, e** The mRNA (**d**; $n = 5$ for CD, $n = 5$ for HFD) and protein (**e**; $n = 4$ per group) expression of B3galt5 in colon of mice fed with chow diet and high-fat diet for 12 weeks, β-Tubulin was used as the loading control. **f, g** The mRNA (**f**; $n = 3$ for WT, $n = 5$ for ob/ob) and protein (**g**; $n = 4$ per group) expression levels of B3galt5 in colon of wild-type mice and ob/ob mice, β-Tubulin was used as the loading control. **h, i** The mRNA (**h**; $n = 21$ for Healthy, $n = 28$ for Obese) and protein (**i**; $n = 6$ per group) expression levels of B3galt5 in colon sections collected from healthy individuals and obese patients, β-Tubulin was used as the loading control. **j** The Spearman's correlation coefficient by two-sided statistical tests, $p$ value, and linear relationship of BMI and the relative mRNA expression of B3galt5 ($n = 48$). Hypo hypothalamus, Jej jejunum, Duo duodenum, eWAT epididymal white adipose tissue, iWAT inguinal white adipose tissue, BAT brown adipose tissue, WT wild type, KO B3galt5 whole-body knockout, CD chow diet, HFD high-fat diet, BMI body mass index. Data are mean ± SEM. Significance was determined using unpaired two-tailed Student's t test (Fig. 3d-i). Source data are provided as a Source Data file.

deficiency leads to mucin being a more susceptibility to bacterial-derived proteolytic degradation, further impairing intestinal permeability, which may contribute to the development of obesity.

Previous studies have reported that B3galt5 is able to catalyze the synthesis of an extended chain of core 3 O-glycans, which is important for maintaining the stability and integrity of the colonic mucosal barrier[36,37]. Thus, we asked whether B3galt5 deficiency destabilizes the mucus layer because of inadequate O-glycosylation of mucin. The

released O-glycans from WT and *B3galt5*[-/-] mice was analyzed using LC-MS[38]. The observed *O*-glycans are summarized in Supplementary Table 1, revealing a significant decrease of core 3 and core 4 *O*-glycans in *B3galt5*[-/-] mice as compared with WT mice (Table S1). The base peak chromatograms of mucus *O*-glycans from WT and *B3galt5*[-/-] mice are shown in Fig. 6f. The major peaks in the LC/MS chromatogram of WT-derived glycans, especially core 3 and core 4 *O*-glycans, such as Fuc-Gal-GlcNAc-GalNAc (733.29a), NeuAc-Gal-GlcNAc-GalNAc (878.33a),

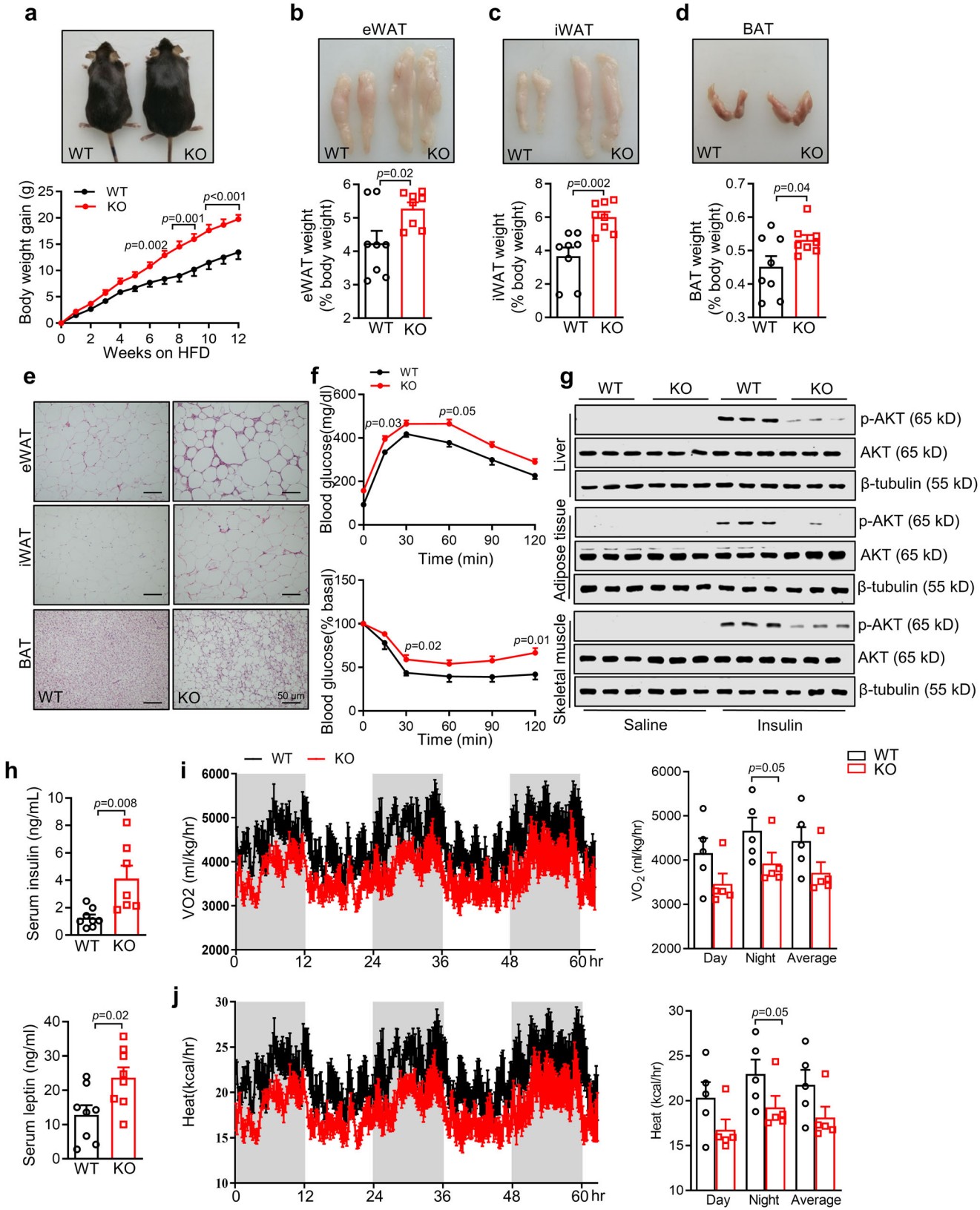

NeuAc-(HexNAc-Gal-GlcNAc-GalNAc) (1081.41), GalNAc-(GlcNAc-Gal-Fuc)$_2$ (1244.48a) and NeuAc-Gal-GlcNAc-GalNAc-GlcNAc-Gal-Fuc (1389.519), were almost reduced or even disappeared in *B3galt5*$^{-/-}$ mice-derived glycans (Fig. 6f). Moreover, consistent with the disordered mucus *O*-glycosylation, HFD-fed *B3galt5*$^{-/-}$ mice showed elevated serum endotoxin levels from portal vein and proinflammatory cytokines upregulation in the intestine (Fig. 6g, h). Furthermore, TBC treatment increased the thickness of mucus layer in HFD-induced mice (Fig. 6i) but failed to elevate the thickness of mucus layer in PXR$^{-/-}$ mice (Figure S7e), which may help explain why TBC administration alleviates obesity and insulin resistance. Together, these results indicate that B3galt5 regulates obesity by influencing mucin *O*-glycosylation;

**Fig. 4 | B3galt5 whole-body knockout exacerbates high-fat diet-induced obesity and insulin resistance in mice.** Wild-type (WT) mice and B3galt5 whole-body knockout (KO) mice were fed with a high-fat diet (HFD) for 12 weeks. **a** Appearance (top) and growth curve (bottom) of WT and B3galt5 KO mice (*n* = 13 for WT, *n* = 14 for KO). (**b-d**) Representative photographs and the ratio of fat depots to body weight of eWAT (**b**, top), iWAT (**c**, top), and BAT (**d**, top) (*n* = 8 per group). (**e**) H&E staining of adipose tissues of eWAT, iWAT and BAT. Scale bar: 50 μm. **f** Blood glucose concentrations during GTT (2 g/kg; top) and ITT (0.75 U/kg; bottom) in WT and B3galt5 KO mice (*n* = 8 per group). **g** Protein levels of phosphorylated Akt and total Akt in the liver, adipose tissue, and skeletal muscle after insulin injection. HFD-fed mice received a bolus injection of insulin (1 U/kg) through the portal vein,

β-Tubulin was used as the loading control. (**h**) Serum insulin (*n* = 8 for WT, *n* = 7 for KO) and leptin (*n* = 8 per group) levels in WT and B3galt5 KO mice. **i, j** Oxygen consumption (**i**) and whole-body energy expenditure (**j**) of WT and B3galt5 KO mice (*n* = 5 per group). eWAT epididymal white adipose tissue, iWAT inguinal white adipose tissue, BAT brown adipose tissue. Data are mean ± SEM. All the data were assessed form normal distribution before statistical analysis. Significance was analyzed using two-way analysis of variance (ANOVA) with Sidak's multiple comparisons test (Fig. 2a and f). The remaining statistical differences were determined using unpaired two-tailed Student's t test. Source data are provided as a Source Data file.

B3galt5 deficiency disrupts mucin *O*-glycosylation, sensitizes mucus for proteolytic degradation, thus impairing the intestinal permeability.

B3galt5 is also responsible for the galactosylation of GlcNAc-based acceptors and glycosphingolipids (GSLs), which are known to be involved in the pathogenesis of insulin resistance in type 2 diabetes[39,40]. Particularly, B3galt5 catalyzes the galactosylation of glycosphingolipid globoside-4 (Gb4) to form globoside-5 (Gb5, also named SSEA3)[41,42]. Therefore, we also investigated whether B3galt5 deficiency influenced the levels of GSLs in mice. Surprisingly, lipidomics analysis targeted GSLs showed that WT and *B3galt5⁻/⁻* mice had comparable levels of GSLs in the intestine and serum, including ceriimide, glucosylceramides, lactosylceramides and Gb3 (Figure S7f-i). Flow cytometry analysis revealed that SSEA3 level in intestine showed no difference between WT and *B3galt5⁻/⁻* mice (Figure S7j-k). These results suggested that B3galt5 may not regulate obesity by altering the levels of GSLs.

### Intestinal specific B3galt5 knockout aggravates HFD-induced obesity, insulin resistance and inflammation

Given that B3galt5 was highly expressed in colon, to further confirm the intestinal B3galt5 function influencing metabolic disorders, we generated intestinal epithelial cell-specific B3galt5 knockout mice (*B3galt5△IEC*) by intercrossing *B3galt5f/f* mice with Villin-Cre-ERT mice (Figure S8a). The protein expression of B3galt5 was found substantially reduced in all sections of intestine in *B3galt5△IEC* mice, comparing to *B3galt5f/f* mice (Figure S8b).

On a 12-weeks chow diet, *B3galt5△IEC* and *B3galt5f/f* mice showed similar body weight, adipose tissue weight and insulin sensitivity (Figure S8c-e). After 12-weeks HFD feeding, *B3galt5△IEC* mice were visibly bigger and gained more body weight than *B3galt5f/f* mice (Fig. 7a, b). Consistent with *B3galt5⁻/⁻* mice, the fat pats including eWAT, iWAT and BAT from *B3galt5△IEC* mice were significantly larger and heavier than those from *B3galt5f/f* mice (Fig. 7c, d; Figure S9a-b). H&E staining and adipocytes area quantity indicated that *B3galt5△IEC* mice had larger adipocytes than *B3galt5f/f* mice in eWAT, iWAT, BAT (Fig. 7e; Figure S9c-d). HFD-fed *B3galt5△IEC* mice showed impaired glucose tolerance and insulin sensitivity (Fig. 7f, g). The deteriorated insulin resistance in *B3galt5△IEC* mice was further evidenced by elevated serum levels of insulin and leptin (Fig. 7h). Besides, *B3galt5△IEC* mice showed upregulated pro-inflammatory cytokines levels and enhanced macrophages infiltration in both the liver (Fig. 7i, j, Figure S9e) and adipose tissue (Fig. 7k, l, Figure S9f). Flow cytometry analysis of ATMs showed slightly increased proportion of M1 macrophages and greatly reduced percentage of M2 macrophages in HFD-fed *B3galt5△IEC* mice (Figure S10a-b). As a result, the ratio of M1 to M2 macrophages was notably increased in eWAT of *B3galt5△IEC* mice comparing to the *B3galt5f/f* mice (Figure S10b). These results indicate that specific ablation of B3galt5 in intestine results in severe HFD-induced obesity, insulin resistance and inflammation as consistent with B3galt5 whole-body knockout.

Moreover, HFD-fed *B3galt5△IEC* mice showed elevated serum endotoxin levels from portal vein (Fig. 7m) and markedly increased serum concentration of FITC-dextran compared with *B3galt5f/f* mice

(Fig. 7n). Further alcian blue staining revealed that *B3galt5△IEC* mice had a thinner colonic mucus layer than *B3galt5f/f* mice (Fig. 7o). As expected, immunofluorescence staining also showed an impaired Muc2-stained inner mucus layer in *B3galt5△IEC* mice (Fig. 7o). The liver injury and hepatic lipid accumulation were also significantly severe in *B3galt5△IEC* mice (Figure S10c-d). These data suggest that the intestinal specific B3galt5 knockout results in undermined intestinal barrier, and thus may contribute to the deterioration of obesity, insulin resistance, and inflammation.

### B3galt5 mediates the beneficial effects of intestinal PXR activation on obese mice

Our results so far have supported that B3galt5 is a direct target gene of intestinal PXR and regulates the progression of obesity. To investigate whether B3galt5 is dispensable for the protective effect of intestinal PXR activation, we fed *B3galt5⁻/⁻* mice with TBC supplemented in HFD for 12 weeks. As expected, the beneficial effect of TBC on obesity was abrogated in *B3galt5⁻/⁻* mice. Treatment with TBC in *B3galt5⁻/⁻* mice did not diminish HFD-induced body weight gain (Fig. 8a), ratio of fat pad weight to body weight, and adipocyte size (Fig. 8b-d). Little improvement of glucose tolerance and insulin sensitivity in TBC-treated *B3galt5⁻/⁻* mice was observed compared to the vehicle-treated counterparts, as assessed by GTT, ITT, and serum levels of leptin and insulin (Fig. 8e-h). Taken together, these results suggest that the beneficial effects of TBC depend on B3galt5, confirming B3galt5 is the messenger conducting intestinal PXR function and the PXR-B3galt5 axis is vital for intestinal barrier maintenance, thus preventing the occurrence of obesity and inflammation.

## Discussion

In this study, we provide the first evidence that selective activation of intestinal PXR can alleviate diet-induced obesity and insulin resistance by upregulating intestinal B3galt5 expression. We identified B3galt5 is a direct transcriptional target of PXR. In whole-body B3galt5 knockout mice, we found aggravated obesity, insulin resistance and tissue inflammation upon HFD challenge. In addition, B3galt5 is highly expressed in colon, and intestinal-specific B3galt5 ablation mice also displayed worse metabolic disorders as well as intestinal barrier integrity with an HFD. Furthermore, we showed that B3galt5 is required for the beneficial effects of intestinal PXR activation on HFD-induced metabolic disorders. Based on these observations, we conclude that intestinal PXR may serve as an important role in metabolic diseases through providing a functional intestinal barrier and B3galt5 is the key executor.

The roles of PXR in glucose and lipid metabolism have been explored in the past two decades. However, the studies of PXR in metabolic disorders exhibited controversial results[43-46]. Treating mice with PCN (50 mg/kg, daily) for 4 days aggravated HFD-induced hepatic steatosis but improved glucose tolerance[47]. In contrast, Ma et al. found that 7 weeks treatment of PCN (50 mg/kg, twice weekly) improved HFD-induced obesity and fatty liver[24]. These conflicting results may be due to the different periods of treating time course. In genetic mouse models, whole-body PXR knockout ameliorated

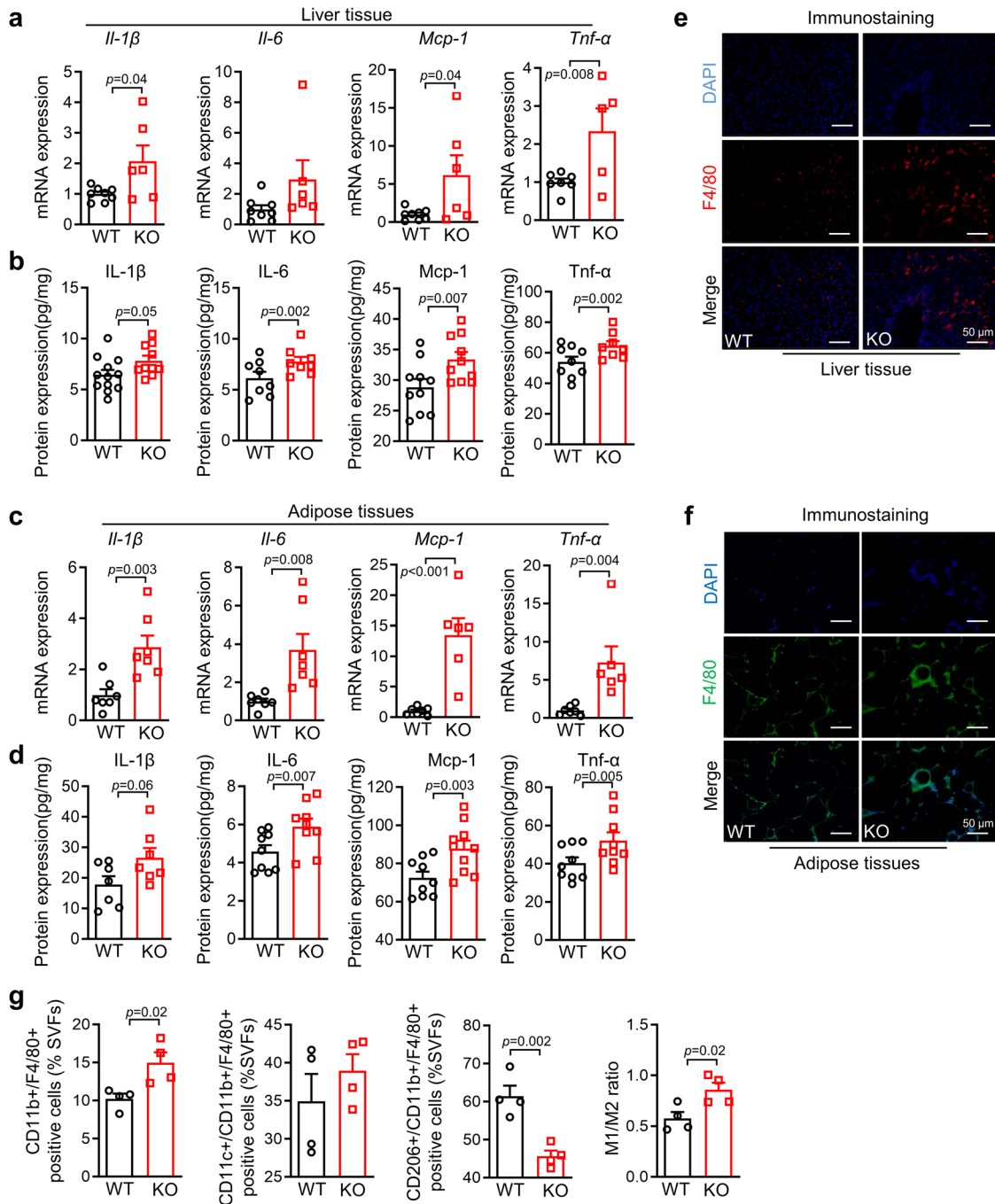

**Fig. 5 | B3galt5 whole-body knockout exacerbated HFD-induced systemic inflammation in mice. a** The mRNA expression of inflammation-related gene *Il-1β*, *Il-6*, *Mcp-1* and *Tnf-α* in liver (*n* = 8 for WT, *n* = 6 for KO). **b** The protein levels of IL-1β, IL-6, Mcp-1 and Tnf-α in liver determined by Elisa (*n* = 12 for WT, *n* = 10 for KO). **c** The mRNA expression of inflammation-related gene *Il-1β*, *Il-6*, *Mcp-1* and *Tnf-α* in adipose tissue (*n* = 7 per group). (**d**) The protein levels of IL-1β, IL-6, Mcp-1 and Tnf-α in adipose tissue determined by Elisa (*n* = 7 per group). **e**, **f** Immunofluorescence staining of F4/80 in the liver (**e**) and adipose tissue (**f**). Scale bar: 50 μm. **g** FACS analysis of M1 and M2 phenotype of macrophages in eWAT of WT and B3galt5 KO mice fed HFD for 12 weeks (*n* = 4 per group). WT wild type, KO B3galt5 whole-body knockout, M1 type 1 macrophages, M2 type 2 macrophages, eWAT epididymal white adipose tissue. Data are mean ± SEM. Significance was determined using unpaired two-tailed Student's t test. Source data are provided as a Source Data file.

diet and genetic induced obesity and insulin resistance[22]. More importantly, hepatic activation of PXR resulted in hypertriglyceridemia, fatty liver and glucose tolerance in *ob/ob* mice[48]. Since PXR is mainly expressed in liver and intestine, these results suggest that PXR may function differently in these tissues. Yet, no study has been conducted to investigate the role of intestinal PXR in metabolic diseases. By using TBC, an intestinal-selective PXR activator, we first provide evidence that selective activation of PXR in the intestine ameliorated HFD-induced obesity and insulin resistance, suggesting a protective role of intestinal PXR in metabolic disorders, which is depart from the deleterious effect of hepatic PXR. The contrary effect of intestinal and liver PXR in obesity might explain the inconsistent results conducted in a whole-body fashion. Selective activation of PXR and its downstream target in the intestine to improve obesity and related metabolic disorders may have more clinical significance with minimal side effects.

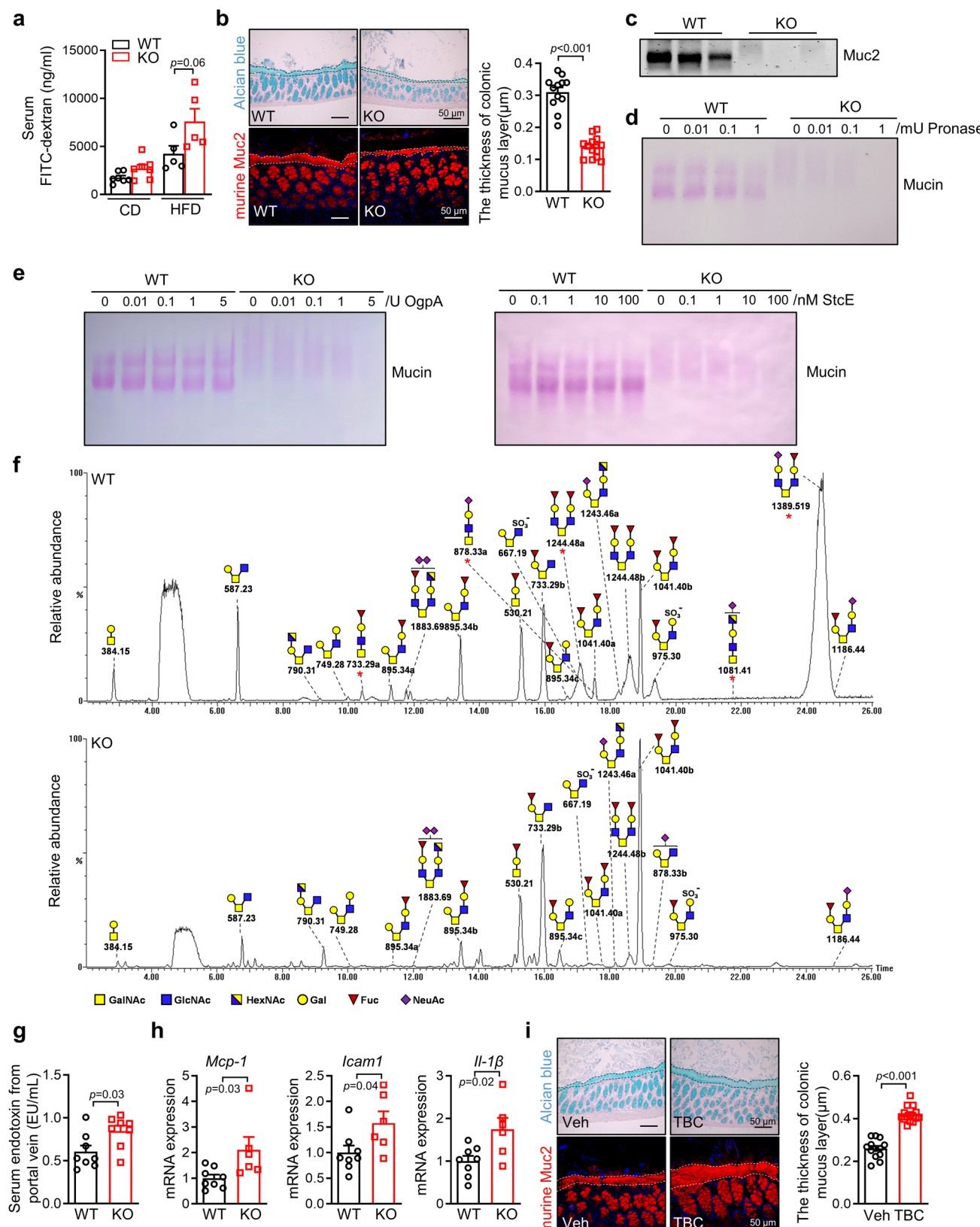

Despite accumulating evidence has reported that PXR activation modulates inflammation in intestinal mucosa barrier[49] or is able to ameliorate dextran sulfate sodium-induced inflammatory disease[50], the targeted genes to execute the beneficial effect of PXR in intestine have not been fully identified. Using RNA-seq data from the intestine of mice treated with PCN, we found B3galt5 was one of the leading candidates

for further evaluation. We found that PXR knockout failed to elevate B3galt5 expression under TBC stimulation, and mutation of B3galt5 promoter disrupted the interaction with PXR. This confirms that B3galt5 is the direct and specific transcriptional target of intestinal PXR.

B3galt5 has been associated with cancer-related proteins' glycosylation due to its ability to catalyzes the galactosylation of

**Fig. 6 | B3galt5 deficiency disrupts mucus *O*-glycosylation and increases permeability of the intestinal barrier. a** The concentration of FITC-dextran in serum of WT and B3galt5 KO mice fed with CD or HFD for 12 weeks ($n = 7$ for CD, $n = 5$ for HFD). **b** Aician blue staining (top) and immunofluorescence staining of Muc2 (bottom) of colon in WT and B3galt5 KO mice fed with 12-week HFD. Quantification of mucus layer thickness (right; $n = 12$ for WT, $n = 14$ for KO). Scale bar: 50 µm. **c** Western blot analysis of Muc2 in intestinal mucus layer of WT and B3galt5 KO mice. **d** PAS staining of mucin treated with pronase in WT and B3galt5 KO mice. **e** PAS staining of mucin treated with OgpA and StcE in WT and B3galt5 KO mice. **f** Negative-ion mode capillary-LC/MS base peak chromatograms of O-glycan alditols of colonic mucin extracted from WT and B3galt5 KO mice. **g** The concentration of endotoxin in portal vein serum of WT and B3galt5 KO mice ($n = 8$ per group). **h** The mRNA expression of inflammation-related gene *Mcp-1*, *Tnf-α* and *Il-1β* in intestine of WT and B3galt5 KO mice fed with 12-week HFD ($n = 8$ for WT, $n = 6$ for KO). **i** Aician blue and Muc2 immunofluorescence staining of colon in WT mice fed with HFD supplemented with TBC 0.05% (w/w) for 12 weeks. Quantification of mucus layer thickness (right; $n = 13$ for Veh, $n = 15$ for TBC). Scale bar: 50 µm. CD chow diet, HFD high-fat diet, WT wild type, KO B3galt5 whole-body knockout, OgpA O-glycoprotease, StcE the secreted protease of C1 esterase inhibitor. Data are mean ± SEM. Significance was determined using unpaired two-tailed Student's t test. Source data are provided as a Source Data file.

glycosphingolipid globoside-4 (Gb4) to form globoside-5 (Gb5), which also known as SSEA3, a common cancer-specific marker highly expressed in breast cancer stem cells[51]. B3galt5 can also catalyze the synthesis of tumor markers CA19-9, which plays an important role in the development of pancreatitis and pancreatic cancer in mice[52]. In our study, we found B3galt5 appeared to be downregulated in the metabolic disorder progression. And the down-regulation of B3galt5 could be a result of decreased expression of PXR. Previous study has showed HFD could impact the expression of PXR and its downstream target gene Cyp3all[53]. High fat diet feeding could significantly decrease the PXR expression in liver. We also found that HFD could also significantly decrease PXR level in colon (not show in this study). In this study, we have confirmed B3galt5 was a target gene of PXR and can be induced by PXR activation. During HFD, the downregulated PXR might decrease B3galt5 expression, which might be assumed in humans. To further test the role of B3galt5 in metabolic diseases, we generated whole-body B3galt5 knockout mice and found exacerbated obesity, insulin resistance, and inflammation when challenged with high-fat diet. Such phenotypes were also observed in the intestine-specific B3galt5 knockout (*B3galt5△IEC*) mice, in accordance with the highly B3galt5 expression in the colon.

In intestine, the mucin is the major structural and functional components in mucus[54] and is often decorated by *O*-glycosylation as well as other complex modifications. Disruption of *O*-glycosylation may destabilize the mucus layer[8], alter the interactions between glycans and lectins[55], the host and intestinal flora[56], and interrupt lymphocyte homing and inflammation[57]. Recent studies have reported that dysfunctional mucus leads to impaired intestinal barrier integrity, which contributes to obesity and other metabolic disorders[58]. To demonstrate why B3galt5 deficiency aggravates obesity and other metabolic disorders, we tested the integrity of the intestinal barrier in B3galt5 knockout mice. In our study, the intestinal permeability was compromised due to the thinner mucus layer shift in *B3galt5−/−* mice, which was confirmed by a previous study showing the decreased mucus layer[59]. We also found B3galt5 deficiency was likely involved in the disruption of *O*-glycosylation from lack of core 3 and core 4 O-glycans. However, we have only focused on *O*-glycosylation, and future studies should explore whether other types of glycosylation are involved in obesity and metabolic disorders.

To test the hypothesis that enhanced activation of intestinal PXR playing a protective role in metabolic disorders is B3galt5-dependent, we administrated B3galt5 knockout mice with TBC. TBC treatment failed to alleviate body weight gain, insulin resistance, and inflammation in *B3galt5−/−* mice, along with the unrecovered mucin layer. Here, we provide intestinal PXR and B3galt5 as promising therapeutic targets for clinical practice. Rifaximin is a human intestinal-selective PXR agonist with broad-spectrum antibacterial activity. Because rifaximin is not easily absorbed into systemic circulation, oral rifaximin can only reach a high drug concentration in the intestinal tract, and therefore possess a high safety with minimal systemic side effects. In clinic, rifaximin is mainly used for the treatment of bacterial intestinal infections, hepatic encephalopathy and inflammatory bowel disease[60].

Recently, rifaximin-α was found to alleviate liver cirrhosis and encephalopathy by reducing gut-derived inflammation and mucin degradation[61]. In our present study, we have confirmed PXR-B3galt5 axis as a therapeutic target in obesity and related metabolic disorders via regulating *O*-glycosylation of mucins. As a clinical medication, whether rifaximin can alleviate obesity and the underlying mechanism needs further investigation.

In summary, our data demonstrated that selectively activation of intestinal PXR alleviated diet induced obesity and insulin resistance by upregulating intestinal B3galt5. We also uncovered a role for B3galt5 as a downstream target gene of PXR in regulating obesity, insulin resistance and systemic inflammation by influencing *O*-glycosylation of colonic mucus. Thus, we have provided proof-of-concept evidence that modulation of intestinal mucus barrier by targeting intestinal PXR or B3galt5 is an attractive strategy for the prevention and therapy of obesity and related metabolic disorders.

## Methods
### Human samples
All patients were recruited from First Affiliated Hospital of Kunming Medical University (Kunming, China). Biopsy samples were harvested from the proximal colon of healthy individuals (18.5 kg/m² ≤ BMI ≤ 24.9 kg/m², $n = 21$) or obese patients (BMI ≥ 28 kg/m², $n = 28$), as describe previously[62]. The baseline characteristics of all subjects are described in Table S2. This study was approved by The Clinical Research Ethics Committee of Kunming Medical University (No:2017L27). Ethical approval and written informed consent were also obtained from all subjects.

### Animals
All animal protocols were approved by Sichuan University Animal Care and Use Committee (No: 20210222030). Mice were housed at 25 °C in a 12-h light-dark cycle in the animal facility at West China Hospital, Sichuan University. Food and water were available ad libitum. *Ob/ob* mice and C57BL/6 J mice were provided by Beijing HFK Bioscience (Beijing, China). PXR knockout (PXR−/−) mice were obtained as a gift from Capital Medical University (Beijing, China). B3galt5 knockout mice (*B3galt5−/−*), B3galt5-floxed (*B3galt5f/f*) mice and VillinCre-ERT mice were purchased from Beijing Biocytogen Pharmaceuticals Co., Ltd (Beijing, China). The guide RNA sequence (5′-3′) of *B3galt5−/−* mice were CCACTCGTTATATATGTGTTTGG, CCAAACACATATATAACGAG TGG, TTTGAACGAGTAAGTGACCCTGG, and TGCTGGCTCTTAACC-TACCC AGG. *B3galt5f/f* mice were interbred with VillinCre-ERT mice to generate intestine-specific B3galt5-deficient mice (*B3galt5△IEC*).

### Luciferase assay
HEK293 cells were seeded in 48-well plates and co-transfected with mouse B3galt5 luciferase reporter or B3galt5 mutant luciferase reporter and mouse PXR plasmid. After 24 hours of transfection, HEK293 cells were treated with mouse PXR selective agonist PCN (10 µM) for another 24 h, and cells were harvested to detect luciferase. β-galactosidase activity was used to normalize transfection efficiency.

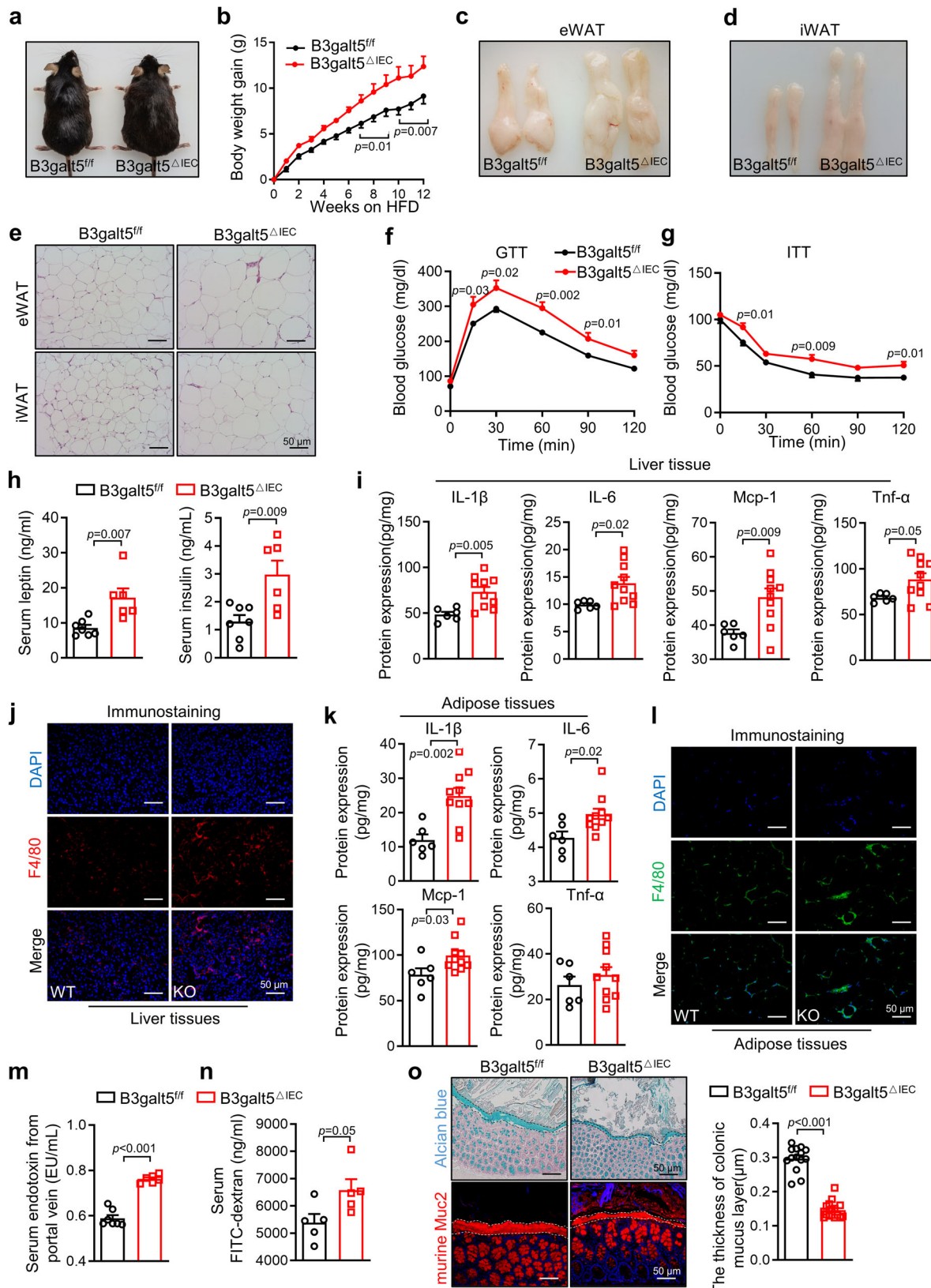

**Chromatin immunoprecipitation (ChIP)**
8-week-old C57BL6/J mice were administrated of a single dose of 200 mg/kg PCN or vehicle (DMAO/corn oil; v/v; 1/3) intraperitoneally. After 10 h treatment, the colon was harvested for ChIP according to the commercial manufacture's guide (Millipore, Billerica, MA). Chromatin samples were incubated with PXR antibody (sc-48240, Santa, USA) at 4°C overnight. Purified DNA was subjected to RT-PCR. Primer sequences are listed in Table S2.

**Biochemical analysis**
Serum triglyceride, glucose and cholesterol concentration were measured using commercial kits (Biosino, Beijing, China). Serum

**Fig. 7 | Intestinal specific B3galt5 knockout accelerates HFD-induced obesity and systemic inflammation.** *B3galt5*^f/f and *B3galt5*^△IEC mice were fed with HFD for 12 weeks. **a**, **b** Appearance (**a**) and growth curve (**b**) of *B3galt5*^f/f and *B3galt5*^△IEC mice (*n* = 7 for *B3galt5*^f/f, *n* = 6 for *B3galt5*^△IEC). **c**, **d** Representative photographs of eWAT (**c**), iWAT (**d**). (**e**) H&E staining of eWAT and iWAT. Scale bar: 50 µm. **f**–**g** Blood glucose concentrations during GTT (1 g/kg; f) and ITT (1.5 U/kg; g) in *B3galt5*^f/f and *B3galt5*^△IEC mice (*n* = 7 for *B3galt5*^f/f, *n* = 6 for *B3galt5*^△IEC). **h** Serum insulin and leptin levels in *B3galt5*^f/f and *B3galt5*^△IEC mice (*n* = 7 for *B3galt5*^f/f, *n* = 6 for *B3galt5*^△IEC). **i** The protein levels of IL-1β, IL-6, Mcp-1 and Tnf-α in liver determined by Elisa (*n* = 6 for *B3galt5*^f/f, *n* = 10 for *B3galt5*^△IEC). **j** Immunofluorescence staining of F4/80 in liver. Scale bar: 50 µm. **k** The protein levels of IL-1β, IL-6, Mcp-1 and Tnf-α in adipose tissue determined by Elisa (*n* = 6 for *B3galt5*^f/f, *n* = 10 for *B3galt5*^△IEC). **l** Immunofluorescence staining of F4/80 in adipose tissue. Scale bar: 50 µm.

**m** Portal vein serum endotoxin level (*n* = 7 for *B3galt5*^f/f, *n* = 6 for *B3galt5*^△IEC). **n** The concentration of FITC-dextran in serum of *B3galt5*^f/f and *B3galt5*^△IEC mice fed with HFD for 12 weeks (*n* = 5 per group). **o** Aician blue staining and immunofluorescence staining of Muc2 in colon. Quantification of mucus layer thickness (right; *n* = 14 for *B3galt5*^f/f, *n* = 16 for *B3galt5*^△IEC). Scale bar: 50 µm. *B3galt5*^f/f B3galt5 floxed mice, *B3galt5*^△IEC intestine-specific B3galt5-deficient mice, eWAT epididymal white adipose tissue, iWAT inguinal white adipose tissue, GTT glucose tolerance test, ITT insulin tolerance test. Data are mean ± SEM. All the data were assessed form normal distribution before statistical analysis. Significance was analyzed using two-way analysis of variance (ANOVA) with Sidak's multiple comparisons test (Fig. 7b, f and g). The remaining statistical differences were determined using unpaired two-tailed Student's t test. Source data are provided as a Source Data file.

insulin or leptin concentrations were determined using a Rat/Mouse Insulin ELISA Kit (EZRMI-13K, Merck Millipore, Darmstadt, Germany), Mouse Leptin ELISA Kit (EZML-82K, Merck Millipore, Darmstadt, Germany), respectively.

## Glucose and insulin tolerance tests
For glucose tolerance test (GTT), mice were fasted for 16 h and intraperitoneally injected with an indicated dosage of glucose. For insulin tolerance test (ITT), the mice were fasted for 4 h and intraperitoneally injected with insulin at indicated dosage. The blood was collected from the tip of the tail vein at predetermined time points and detected by a glucose meter (Roche, Basel, Switzerland).

## Electrophoretic mobility shift assay (EMSA)
EMSA was performed using $^{32}$P-labeled oligonucleotides and receptor proteins prepared by the TNT method[63]. Briefly, $^{32}$P-labeled probes were mixed with the fusion proteins in a binding buffer for 30 min at 25°C. The Cyp3a23 $^{32}$P-labeled probes was used as a positive control.

## Intestinal permeability
FITC-labelled dextran was used to evaluate the in vivo permeability of intestinal barrier[64]. Mice were fasted for 4 h and then given with FITC-labelled dextran (FD4; Sigma-Aldrich) by gavage at a dose of 400 mg per 1 kg body weight. Serum was collected 4 h after administration, and detected for fluorescence intensity (excitation: 492 nm; emission: 525 nm).

## Mucin extraction and PAS analysis
The colonic mucus layer of mice was scraped and homogenized in extraction buffer (6 M GuCl; 5 mM EDTA.2Na; 10 mM NaH2PO4) and then rotated overnight at 4°C. The precipitate was collected by centrifugation at 14000 g for 30 min. Reduction buffer (6 M GuCl; 5 mM EDTA.2Na; 0.1 M Tris-HCl) containing 100 mM DTT was added to the precipitate and incubated for 5 h at 37°C. 250 mM IAA was then added and incubated overnight in the dark. The liquid obtained from the previous steps was dialyzed and lyophilized to obtain purified mucin. For mucin degradation, the purified mucin was dissolved in 50 mM ammonium bicarbonate and incubated with pronase (Millipore, 53702), StcE (Sigma, SAE0202), or OgpA (O-glycoprotease, New England Biolabs, P0761S) for 3 h at 37°C.

Mucin protein was loaded and separated by 0.8% TAE-SDS agarose, then transferred to nitrocellulose membranes. After 1 h of blocking with 0.5% casein, the membranes were incubated with Muc2 antibody. For PAS staining, mucin protein transferred to nitrocellulose membranes was stained with Periodic Acid-Schiff (PAS) according to the directions of PAS kit (Solarbo, G1281).

## Glycan analysis by LC-MS
The mucin glycan analysis by LC-MS was performed as previously described[65]. Mucin O-glycans were released by reductive β-elimination

in 100 µL NaOH (50 mM) with 1 M NaBH$_4$ and then neutralized by adding 10 µL 2 M acetic acid until the bubbling stopped. Samples were dried in vacuo and dissolved in ultrapure H$_2$O and loaded onto the cartridges (Supelco; Bellefonte, PA, USA), followed by ultrapure H$_2$O. Reduced O-glycans were eluted, dried in vacuo and lyophilized. Samples were dissolved in 30 µL ultrapure H$_2$O prior to analysis.

HPLC was performed on an Agilent 1290 Infinity system. LC-MS was conducted in negative ion mode on an Agilent 6530 QToF-MS with an Agilent Jet Stream electrospray ionization source. LC-MS data acquisition and analysis were performed using MassHunter Workstation software (Agilent Technologies): Data Acquisition Workstation (v B.06.01, SP1) and Qualitative Analysis (v B.07.00, SP2).

## Quantitative real-time PCR
Total RNA extracted from tissues or cells and reverse-transcribed into complementary DNA (cDNA) using a PrimeScript™ RT reagent kit (Takara, Kusatsu, Japan). The mRNA was quantified by CFX96 Real-time RT-PCR System with SYBR Green (Bio-Rad, CA, USA). The primer sequences were listed in Table S3. Relative mRNA levels were normalized to that of 18 s by comparison with the control groups using CFX Manager™ software (Bio-Rad, CA, USA).

## Western blot analysis
The total proteins were extracted by loading buffer, applied to SDS-PAGE gels and probed with antibodies. The signal was detected by a LI-COR System (Lincoln, NE) and semi-quantified using Image J software (version 1.8.0, https://imagej.nih.gov/ij/) normalized by β-Tubulin bands. Antibodies were listed in table S4. The uncropped blots were provided in the Source Data file.

## Statistics
All results were presented as means ± SEM of at least three independent experiments. Statistical analysis was performed using GraphPad Prism version 8. All the data were assessed form normal distribution before statistical analysis. Significance was analyzed using two-way analysis of variance (ANOVA) with Sidak's multiple comparisons test (Figs. 1e, k, l, 4a, f, 7b, f, g, 8a and SFig. 4c). In these cases, differences were considered significant at $^{\#}P < 0.05$, $^{\#\#}P < 0.01$. The $n \leq 3$ data sets were analyzed using non-parametric approaches. The remaining statistical differences between groups were determined using unpaired two-tailed Student's t test or one-way ANOVA with post-hoc Tukey test. Spearman's correlation was used to cross correlate BMI and gene expression data using GraphPad Prism version 8. *P < 0.05 and **P < 0.01. P < 0.05 were considered statistically significant. At least three independent experiments were repeated with similar results.

## Reporting summary
Further information on research design is available in the Nature Portfolio Reporting Summary linked to this article.

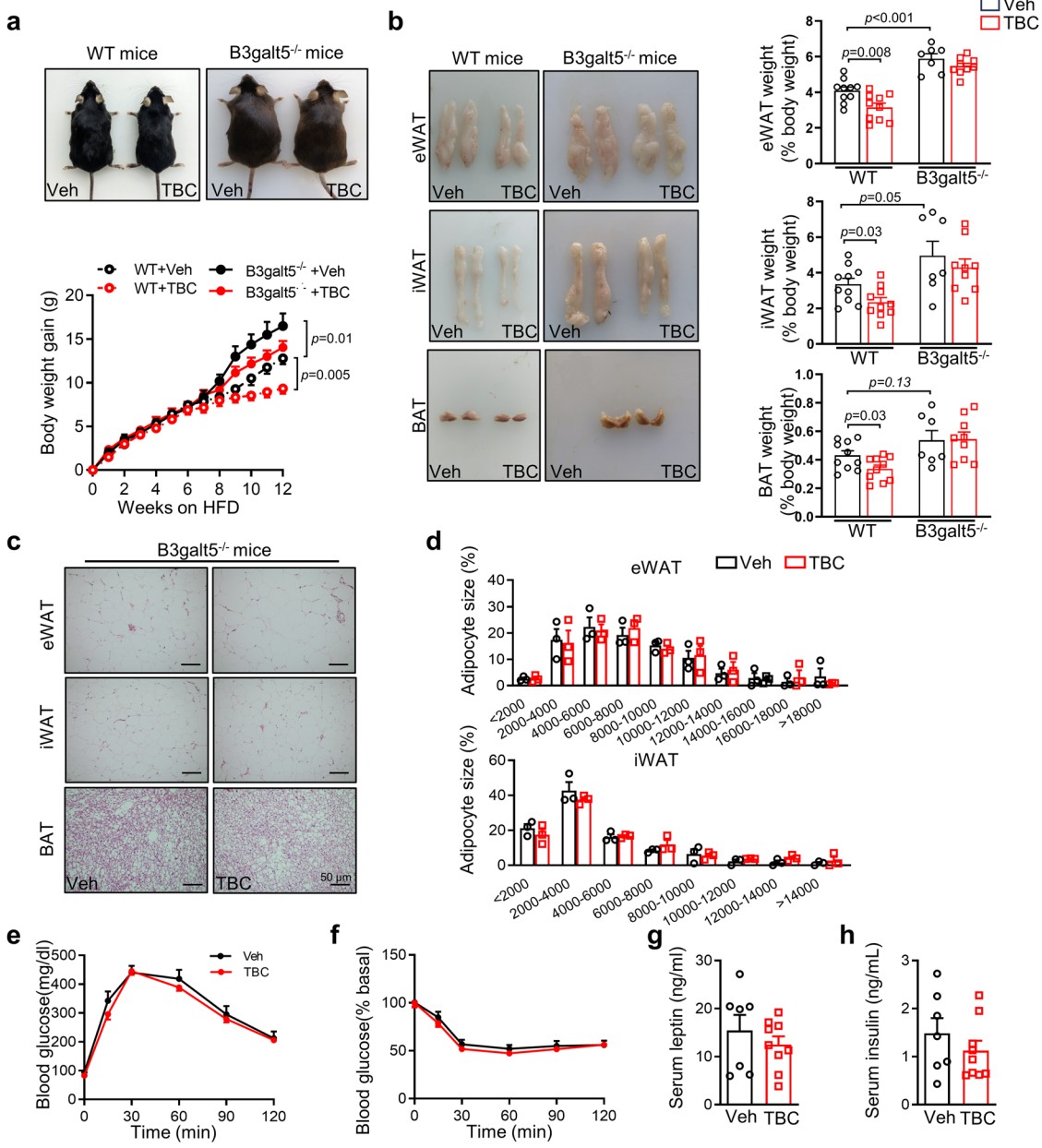

**Fig. 8 | B3galt5 mediates the effects of intestinal of PXR activation on obese mice.** WT and B3galt5 KO mice were fed with TBC supplemented in HFD at 0.05% (w/w) for 12 weeks. **a** Appearance (top) and growth curve (bottom) of vehicle (Veh) and TBC treated WT and B3galt5 KO mice (n = 10 for WT + Veh and WT + TBC, n = 7 for B3galt5-/- + Veh, n = 9 for B3galt5-/- + TBC). **b** Representative photographs of eWAT, iWAT and BAT (left) and the ratio of fat depots to body weight (right; n = 7–10 as (**a**)). **c, d** H&E staining of adipose tissues (**c**) and distribution of adipocyte size of eWAT and iWAT (**d**; n = 3 per group). Scale bar: 50 μm. **e, f** Blood glucose concentrations during GTT (2 g/kg; **e**) and ITT (1 U/kg; **f**) in Veh and TBC-treated B3galt5 KO mice (n = 7 for B3galt5-/- + Veh, n = 9 for B3galt5-/- + TBC). **g** Serum

leptin levels in Veh and TBC treated B3galt5 KO mice (n = 7–9 as (**e-f**)). **h** Serum insulin levels in Veh and TBC treated B3galt5 KO mice (n = 7–9 as (**e–f**)). WT wild type; B3galt5-/-: B3galt5 whole-body knockout, TBC tributyl citrate, eWAT epididymal white adipose tissue, iWAT inguinal white adipose tissue, BAT brown adipose tissue. Data are mean ± SEM. All the data were assessed form normal distribution before statistical analysis. Significance was analyzed using two-way analysis of variance (ANOVA) with Sidak's multiple comparisons test (Fig. 8a). The remaining statistical differences were determined using unpaired two-tailed Student's t test. Source data are provided as a Source Data file.

## Data availability

The RNA-seq datasets generated in this study have been deposited in GEO database under accession code GSE266942 and GSE270696. The remaining data generated in this study are available in the main text or the supplementary materials. Source data are provided with this paper.

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

## Acknowledgements

This work was supported by the National Natural Science Foundation of China (grant number 82025007 to J. H., 81930020 to J. H., 82170874 to Y. L., and 82270923 to L. M.), Innovation Group Project from Science & Technology Department of Sichuan Province (grant number 23NSFTD0067 to J. H.), and National Clinical Research Center for Geriatrics, West China Hospital, Sichuan University (grant number Z2024YY003 to J.H.). The authors thank Miss Li Li and Miss Fei Chen from Institute of Clinical Pathology, West China Hospital of Sichuan University, and Miss Yan Wang from Core Facility of West China Hospital, Sichuan University for technical assistance.

## Author contributions

J.Z., Y.H., Y.L. and J.H. designed the experiments; J.Z. and Y.H. performed experiments. H.L., P.X., Q.L., Z.Z., T.W., Q.T., Q.J., Ying Xu, Yan Xia, X.J., and J.L. helped with experiments. Y.S. provided the human samples. L.M., and W.X contributed to the discussion and reviewed the manuscript. A.Q. provided the PXR knockout mice. J.Z., Y.L. and J.H. wrote the manuscript. Y.L. and J.H. obtained funding. J.Z., Y.H., Y.L. and J.H. are the guarantors of this work and as such, had full access to all the data in the study and take responsibility for the integrity of the data and the accuracy of the data analysis.

## Competing interests

The authors declare no competing interests.
