## [Peer Review File · Nature Communications]

B3galt5 functions as a PXR target gene and regulates obesity and insulin resistance by maintaining intestinal integrityREVIEWER COMMENTS

Reviewer #1 (Remarks to the Author):

In this manuscript, the authors show that B3galt5 regulates HFD-induced obesity by reducing intestinal permeability. The intestine-specific PXR agonist TBC reduced HFD-induced obesity and glucose resistance, in WT mice but not in PXR KO mice. The PXR agonists increased the expression of B3galt5 gene in a PXR-dependent manner. B3galt5 KO and intestine-specific B3galt5 KO mice showed HFD-induced obesity and glucose resistance. B3galt5 KO mice showed a thinner mucus layer in the colon and increased intestinal permeability.

Comments:

1. The authors would like to claim that HFD reduces intestinal barrier function through the reduction of B3galt5 expression and induces obesity. In this case, the authors should provide evidence showing that HFD increases intestinal permeability by reducing the thickness of the mucus layer.
2. beta3galt5 should be B3galt5.
3. The authors should be aware that a previous paper reported a similar intestinal phenotype (Fig. 6 data). Mucosal Immunol 2023 Oct;16(5):624-641. But they ignore it.
4. In Fig. 2a RNA-seq. At which time point, samples were prepared? Are those samples whole intestine? Epithelial cells? Small intestine or large intestine?
5. In Fig. S4a, the authors stated that they generated B3galt5 KO mice by CRISPR-Cas9 system. However, the figure shows only a targeting vector for homologous recombination. The seq of guide RNA should be shown.
6. In Fig. 6b, i, Muc2 staining is in a poor quality. Does the dot line indicate the inner mucus layer? Clearer photos are preferable.

Reviewer #2 (Remarks to the Author):

The authors provide evidence that PXR activation by TBC improved HFD-induced obesity and insulin resistance, likely through an upregulation of the β 3galt5 enzyme expressed in the intestinal mucosa. The β 3galt5 enzyme is required to maintain gut barrier and mucus

integrity. The authors therefore propose the gut-specific activation of PXR as a novel potential therapeutic target in obesity and related metabolic diseases. This is a highly interesting topic with potential clinical impact; however, some important questions remain open and some methodological issues need to be clarified. Without that the work does not sufficiently support the conclusions and claims made in the manuscript.

Major

How were PXR agonists identified? Which ones were tested apart from TBC? Did they yield homogenous results? What about other glycosyltransferases possibly affected by PXR activation? Which role does the animal model used in the study have? What about other models in which metabolic disease was induced by sugars like fructose instead of HFD?

The authors describe that gut barrier is affected by down-regulation of the β 3galt5 enzyme. However, functional evidence is lacking. LPS translocation and liver inflammation/steatosis needs to be evaluated.

How does HFD down-regulate the β 3galt5 enzyme? Which mechanisms can be assumed in humans?

L69f. The authors state that HFD induces gut barrier impairment and mucus disruption. This is correct, although the protagonists of such findings are not cited. Moreover, the role of dietetic sugars, namely fructose, and hyperglycemia, is neglected (see for example ref. 5 and other refs such doi: 10.3945/jn.116.242859).

L77f. Not only positive and protective effects of PXR activation have been described (doi: 10.1016/j.molmet.2023.101779, doi: 10.1016/j.bcp.2021.114698, and many more articles). Which implications do such findings have for the present work? This requires a much more detailed discussion in the manuscript.

L152f. and Fig 3. If β 3galt5 is highly expressed in colon and barely expressed in other tissues including small intestine, one could assume that in particular colonic barrier function is

impaired, and not small intestinal barrier function. Is this correct? And is this in contrast to previous findings derived from mouse feeding experiments?

Fig. 4. Body weight and fat tissues are examined but not liver steatosis. If colonic barrier is disturbed, bacterial translocation into the portal vein and liver can be expected. This needs to be studied and included into the paper to clarify the mechanisms of the PXR-mediated effects.

L165. What means “possibly as a mechanism in PXR dependent metabolic disorders”?

Fig 3 h-i: More clinical data on the human population is required. Most of the patients were only overweight but not obese. Which ones of Table S2 were selected for Fig. 3.? Did e.g. BMI correlate with β 3galt5 expression in colon?

Minor:

L52 ... which leads to increased intestinal permeability and the transference of microbial molecules into systemic circulation.

Into the portal vein blood and liver rather than into systemic circulation. The liver is a clearance organ for most microbial molecules.

L84-85: This sentence is not logical.

Reviewer #3 (Remarks to the Author):

To the authors,

In the current submission, the authors seek to characterize a functional link between the pregnane X receptor (PXR), within the intestine, the expression of B3galt5 and the regulation of the mucous layer in the context of HFD and metabolic dysfunction.

While aspects of the data presented are very interesting, this reviewer has a series of concerns that are outlined below:

Major concerns:

Results - the authors present mucin/mucous layer images throughout the manuscript. The thickness of this layer must be quantified in each set of experiments to support the authors' conclusions.

Line 103 - "in intestine" should read "in the intestine"

Cyp3a11 is predominantly expressed in the small intestine, with very minimal colonic expression. Where in the intestine was the characterization of PXR activation and Cyp3a11 expression assessed?

Figure 1J - given the non-specificity of the B3galt5 antibody (as depicted in Figure 3b, with a dominant "non-specific" band as interpreted by the author that is not apparent in Figure 1J), the authors should present uncropped blots with the appropriate labelling.

Figure 3B - the authors need to include tissues from the B3galt5 KO mice in these blots to justify the interpretation/labelling the "non-specific" band. This also goes for the blots depicted in panels e and g. This reviewer is concerned that the authors are "picking and choosing" a band for quantification without demonstrating the specificity of the antibody. Given that assessing the expression of the B3galt5 is a key metric/outcome in this paper, these blots need to be redone to contain the appropriate controls to show antibody specificity.

Figure 3i - the given the questionable specificity of the B3galt5 antibodies throughout the manuscript, the entire blot (i.e., uncropped) should be presented here.

Figure 4a - this data in this panel are not analyzed correctly. These data should be reanalyzed with the appropriate 2-way ANOVA and post-hoc test. The same reanalysis is

required for panel f.

Figure 5G - Raw flow cytometry plots indicating gating strategy need to be presented here to allow the reader to assess the quality of the data. Furthermore, absolute counts must accompany the cell percentages presented here. Gating strategy and complete methods for tissue dissociation and isolation must also be included in the methods section. As this section reads currently, it would allow a reader to replicate these studies.

Figure 6c - Having seen blots from leading experts in mucin biology over the course of the past 15 years, I can say that this blot is suspiciously clean. To rebut my concerns, the authors should present the entire blot. Furthermore, how can there be so little Muc2 protein in these Bgalt5^{-/-} samples when there is clearly a reasonable signal in both the Alcian blue and Muc2 immunostaining? This reviewer is seriously concerned about the western blot data presented here.

Figure 6D and commentary lines 236-238 - pronase is usually isolated and purified from bacteria that are not gut commensals. Given that there are likely different proteases at play in the gut, the authors must show that either distinct protease known to be expressed by gut bacteria or proteases isolated from gut contents can degrade mucous isolated from WT and B3galt5^{-/-} mice. Without this, the authors cannot make the statements in lines 236-238.

Figure 6i and lines 253-255 - to illustrate the PXR-dependent mechanism of TBC's thickening of the mucus layer, the authors must perform complementary experiments in PXR^{-/-} mice.

Figure 7b-f-g - this data in these panels are not analyzed correctly. These data should be reanalyzed with the appropriate 2-way ANOVA and post-hoc test.

Figure 7 J-L - why were the complementary flow cytometric analyses performed in these experiments? The authors must conduct additional experiments to present quantitative flow cytometry data assessing macrophages in these tissues.

Lines 293-295 - if the authors are going to conclude that intestinal specific B3galt5 KO

results in a compromised barrier, complementary permeability experiments are required (as presented in previous figures).

Figure 8 - the authors need to include WT mice treated with both vehicle and TBC in this study to show that the reagents used in this experiment were actually functional. My first interpretation of the data presented here was that the TBC could just be non-functional in this study. Key controls would counter that conclusion.

Lines 305-308 - without including some key experiments using PXR^{-/-} mice, the authors really can't make the statement that there exists a PXR-B3galt5 axis in this system. Sure, they demonstrate to a certain degree that TBC activates the PXR, but they must perform reciprocal experiments (show that the absence of the PXR leads to the loss of B3galt5 induction in vivo).

Methods - Luciferase Assay - PCN is a mouse selective agonist and shows little activity at the human PXR. Given that HEK293 cells are of human origin, how are the authors able to assess B3galt5 promoter activity using this experimental design? This reviewer is very concerned about these data.

Methods - Statistics - were the data assessed for normal distribution prior to statistical analysis with parametric tests? If so, this needs to be included. If not, non-parametric tests should be employed. Furthermore, N = 3 data sets can not be accurately assessed for normal distribution and should, by default, be analyzed using non-parametric approaches. The data presented in this submission should be reanalyzed accordingly.

Figure 2 and RNA sequencing - what regions of the "intestine" were assessed? As mentioned in a previous comment, Cyp3a11 is minimally expressed in the colon. The methods section should be revised, along with all references to the "intestine" to be more clear as to what regions are being assessed throughout the manuscript.

Minor concerns:

Line 82 - "contribute" should read "contributes".

Lines 101-102 and throughout the manuscript - as a pharmacologist, this reviewer would never use the term "specific" to describe any chemical reagents. The authors should revise this to read "selective". "Specific" can be used in the case of antibodies, but never in the case of chemical reagents since they nearly all show off-target effects (whether they've been reported or not).

Methods for endotoxin and cytokine/chemokine analysis are missing from main body of the manuscript and supplementary methods, as are descriptions of pronase source and use in mucin degradation studies.

Methods - Animals - please include a reference to the approved animal use protocol number here to enhance post-approval assurances.

There is no indication of the breeding scheme and whether littermate controls were used throughout the studies. This is an absolute must when performing studies on intestinal biology and metabolic disease given the microbiota's involvement in regulating these systems. If littermate controls were not used, this is a flaw that constitutes rejection of the paper.

Answers to Reviewers' Comments

Reviewer #1

In this manuscript, the authors show that B3galt5 regulates HFD-induced obesity by reducing intestinal permeability. The intestine-specific PXR agonist TBC reduced HFD-induced obesity and glucose resistance, in WT mice but not in PXR KO mice. The PXR agonists increased the expression of B3galt5 gene in a PXR-dependent manner. B3galt5 KO and intestine-specific B3galt5 KO mice showed HFD-induced obesity and glucose resistance. B3galt5 KO mice showed a thinner mucus layer in the colon and increased intestinal permeability.

1. The authors would like to claim that HFD reduces intestinal barrier function through the reduction of B3galt5 expression and induces obesity. In this case, the authors should provide evidence showing that HFD increases intestinal permeability by reducing the thickness of the mucus layer.

Response: Thanks for the reviewer's constructive comments. Previous studies have demonstrated that HFD could reduce the intestinal mucus layer and increase intestinal permeability¹. Mucin supplement was found can reverse the increased intestinal permeability induced by HFD^{2, 3}. These findings fully confirm that HFD increases intestinal permeability via reducing the thickness of mucus layer. We also detected the increased intestinal permeability by FITC-dextran gavage after HFD feeding (Figure R1a). The Muc2 immunofluorescence staining also showed the reduced thickness of the mucus layer after HFD challenge (Figure R1b), which was consistent with previous studies^{2, 3}. Combined with previous studies, these results suggested that HFD increases intestinal permeability by reducing the thickness of the mucus layer.

Response Figure 1. HFD impaired intestinal barrier and thickened mucin layer. (a) The concentration of FITC-dextran in serum of WT mice fed with CD or HFD for 12 weeks. (b) Immunofluorescence staining of Muc2 of colon in WT mice fed with 12-week HFD. Data are mean \pm SEM, $n = 7$, * $P < 0.05$.

2. beta3galt5 should be B3galt5.

Response: As suggested by the reviewer, we have carefully checked and corrected all the description mistakes.

3. The authors should be aware that a previous paper reported a similar intestinal phenotype (Fig. 6 data). Mucosal Immunol 2023 Oct;16(5):624-641. But they ignore it.

Response: Thanks for the reviewer's comment. We have cited this article (Ref 59) and added the following discussion: In our study, the intestinal permeability was compromised due to the thinner mucus layer in $B3galt5^{-/-}$ mice, which was confirmed by a previous study showing the decreased mucus layer⁴.

4. In Fig. 2a RNA-seq. At which time point, samples were prepared? Are those samples whole intestine? Epithelial cells? Small intestine or large intestine?

Response: Thanks for the reviewer's comments. We have added a comprehensive description in the Supplementary Methods section of our revised manuscript as the following: 8-week-old mice were injected intraperitoneally with 40 mg/kg prenenolone-16 α -carbonitrile (PCN; in DMSO/corn oil 1:3, Cayman Chemical Company, cat # 16343) every 8 h for three times. Mice were sacrificed 8 h after the last administration, and both the ileum and colon tissues were collected for RNA sequencing according to the manufacture's instruction.

5. In Fig. S4a, the authors stated that they generated $B3galt5$ KO mice by CRISPR-Cas9 system. However, the figure shows only a targeting vector for homologous recombination. The seq of guide RNA should be shown.

Response: Thanks for the reviewer's comments. As suggested, we have added the sequence of guide RNA in the Method (Line 406-408): The guide RNA sequence (5'-3') of $B3galt5^{-/-}$ mice were CCACTCGTTATATATGTGTTTGG, CCAAACACATATATAACGAGTGG, TTTGAACGAGTAAGTGACCCTGG, and TGCTGGCTCTTAACCTACCC AGG.

6. In Fig. 6b, i, Muc2 staining is in a poor quality. Does the dot line indicate the inner mucus layer? Clearer photos are preferable.

Response: Thanks for the reviewer's constructive comments. The dot line indicated the inner mucus layer. As suggested by the reviewer, we have also replaced higher quality images of immunofluorescence staining in **Figure 6b** and **6i**, in which the inner mucus layer is more visible.

Reviewer #2

The authors provide evidence that PXR activation by TBC improved HFD-induced obesity and insulin resistance, likely through an upregulation of the $\beta3galt5$ enzyme expressed in the intestinal mucosa. The $\beta3galt5$ enzyme is required to maintain gut barrier and mucus integrity. The authors therefore propose the gut-specific activation of PXR as a novel potential therapeutic target in obesity and related metabolic diseases. This is a highly interesting topic with potential clinical impact; however, some important questions remain open and some methodological issues need to be clarified. Without that the work does not sufficiently support the conclusions and claims made in the manuscript.

1. How were PXR agonists identified? Which ones were tested apart from TBC? Did they yield homogenous results? What about other glycosyltransferases possibly affected by PXR activation? Which role does the animal model used in the study have? What about other models

in which metabolic disease was induced by sugars like fructose instead of HFD?

Response: Thanks for the reviewer's comments. (1) The PXR agonists (PCN, TBC, RIF) used in this study were canonical PXR agonists which was confirmed by numerous studies⁵⁻⁷. At the same time, we detected the expression of Cyp3a11 or MDR1, known target genes of PXR, to confirm the activation of PXR after agonists treatment, as shown in **Figure 2a and 2i**. (2) Apart from TBC, we also used PCN (Pregnenolone 16 α -carbonitrile, a selective mouse PXR agonist) and RIF (rifampicin, a selective human PXR agonist), in Figure 2. (3) The expression of other glycosyltransferases showed no changes in mice colon treated with two canonical PXR agonist, PCN and TBC, in **Figure S2h-i**. (4) The animal models used in the study is high fat diet (HFD)-induced obesity mice, which is an obese animal model confirmed by previous studies⁸⁻¹⁰. (5) In a separate project, we also detected higher expression of B3galt5 after high fructose- and high cholesterol (HFHC) challenge. In HFHC-induced animal model, mice were fed a 40% fat and 0.2% cholesterol diet and given access to fructose-supplemented water (23.9g/L). As showed in **Figure R2**, the expression of B3galt5 was also significantly decreased in HFHC-induced animal models.

Response Figure 2. High fructose- and high cholesterol- diet (HFHC) decreased the expression of B3galt5. The B3galt5 mRNA (a) and protein (b) expression in colon of WT mice fed with CD or HFHC for 16 weeks. Data are mean \pm SEM, n = 7, * P < 0.05.

2. The authors describe that gut barrier is affected by down-regulation of the β 3galt5 enzyme. However, functional evidence is lacking. LPS translocation and liver inflammation/steatosis needs to be evaluated.

Response: Thanks for the reviewer's constructive comments. As suggested, we have detected the serum level of endotoxin and hepatic steatosis in B3galt5^{-/-} and B3galt5 ^{Δ IEC} mice in the revised manuscript. As shown in **Figure 6g and 7m**, the serum LPS level was significantly increased in B3galt5^{-/-} and B3galt5 ^{Δ IEC} mice comparing with their counterparts. Liver inflammation was also elevated in B3galt5^{-/-} and B3galt5 ^{Δ IEC} mice, as evidenced by increased mRNA and protein levels of hepatic inflammatory cytokines (**Figure 5a, 5b, 5e and Figure 7i-j**). Furthermore, B3galt5^{-/-} and B3galt5 ^{Δ IEC} mice displayed more severe liver steatosis as shown by elevated liver injury and increased hepatic lipid deposition (**Figure S7a-b and Figure S10c-d**).

3. How does HFD down-regulate the β 3galt5 enzyme? Which mechanisms can be assumed in

humans?

Response: Thanks for the reviewer's interesting comments. We speculated that down-regulation of B3galt5 seen in obese mouse models could be a result of decreased expression of PXR. Previous study has showed HFD could impact the expression of PXR and its downstream target gene Cyp3a11¹¹. High fat diet feeding could significantly decrease the PXR expression in liver¹². We also found that HFD could also significantly decrease PXR level in colon as shown in **Figure R3**. In this study, we have confirmed PXR-B3galt5 axis is vital for intestinal barrier, thus preventing obesity. B3galt5 was confirmed as a novel target gene of PXR and can be induced by PXR activation. During HFD, the downregulated PXR might decrease B3galt5 expression, which might be assumed in humans.

Response Figure 3. High fat-diet (HFD) decreased the expression of colonic PXR. The B3galt5 mRNA expression in colon of WT mice fed with CD or HFD for 12 weeks. Data are mean \pm SEM, n = 7, * $P < 0.05$.

4. L69f. The authors state that HFD induces gut barrier impairment and mucus disruption. This is correct, although the protagonists of such findings are not cited. Moreover, the role of dietetic sugars, namely fructose, and hyperglycemia, is neglected (see for example ref. 5 and other refs such doi: 10.3945/jn.116.242859).

Response: Thanks for the reviewer's comments. We have added the references about HFD induced gut barrier impairment and mucus disruption (Ref 13-15). As suggested by the reviewer, we have also added the following description in the Introduction and cited relevant references as follows (Line 51-52): High fructose intake induced a loss of mucus thickness in the colon and impaired gut permeability¹³.

5. L77f. Not only positive and protective effects of PXR activation have been described (doi: 10.1016/j.molmet.2023.101779, doi: 10.1016/j.bcp.2021.114698, and many more articles). Which implications do such findings have for the present work? This requires a much more detailed discussion in the manuscript.

Response: Thanks for the reviewer's constructive comments. As suggested, we have engaged in a comprehensive discussion of the intricate function of PXR in conjunction with our present work as follows (Line 319-334): The roles of PXR in glucose and lipid metabolism have been explored in the past two decades. However, the studies of PXR in metabolic disorders exhibited controversial results¹⁴⁻¹⁷. Treating mice with PCN (50 mg/kg, daily) for 4 days aggravated HFD-induced hepatic steatosis but improved glucose tolerance¹¹. In contrast, Ma *et al.* found that 7 weeks treatment of PCN (50 mg/kg, twice weekly) improved HFD-induced obesity and fatty liver²⁰. These conflicting results may be due to the different periods of treating time course.

In genetic mouse models, whole-body PXR knockout ameliorated diet and genetic induced obesity and insulin resistance^{18, 19}. More importantly, hepatic activation of PXR resulted in hypertriglyceridemia, fatty liver and glucose intolerance in *ob/ob* mice¹⁸. Since PXR is mainly expressed in liver and intestine, these results suggest that PXR may function differently in these tissues. Yet, no study has been conducted to investigate the role of intestinal PXR in metabolic diseases. By using TBC, an intestinal-selective PXR activator, we first provide evidence that selective activation of PXR in intestine ameliorated HFD-induced obesity and insulin resistance, suggesting a protective role of intestinal PXR in metabolic disorders, which is depart from the deleterious effect of hepatic PXR. The contrary effect of intestinal and liver PXR in obesity might explain the inconsistent results conducted in a whole-body fashion. Selective activation of PXR and its downstream target in the intestine to improve obesity and related metabolic disorders may have more clinical significance with minimal side effects.

6. L152f. and Fig 3. If β 3galt5 is highly expressed in colon and barely expressed in other tissues including small intestine, one could assume that in particular colonic barrier function is impaired, and not small intestinal barrier function. Is this correct? And is this in contrast to previous findings derived from mouse feeding experiments?

Response: Thanks for the reviewer's comments. Previous studies have suggested colon was the main site of intestinal microbiota colonization and played an important role in maintenance of intestinal barrier²²⁻²⁴. Since B3galt5 is highly expressed in colon, we mainly detected the mucus layer in colon via Alcian blue and Muc2 immunostaining. In this study, we provided evidence that B3galt5 knockout could impair intestinal permeability via thickening colonic mucus layer and disrupting colonic mucus *O*-glycosylation. We will further investigate the small intestinal barrier function in B3galt5 knockout mice.

7. Fig. 4. Body weight and fat tissues are examined but not liver steatosis. If colonic barrier is disturbed, bacterial translocation into the portal vein and liver can be expected. This needs to be studied and included into the paper to clarify the mechanisms of the PXR-mediated effects.

Response: Thanks for the reviewer's constructive suggestions. We have detected the serum level of endotoxin and liver steatosis in B3galt5^{-/-} and B3galt5^{ΔIEC} mice in the revised manuscript. As shown in **Figure 6g and 7m**, the serum LPS level was significantly increased in B3galt5^{-/-} and B3galt5^{ΔIEC} mice comparing with their counterparts. Liver inflammation was significantly elevated in B3galt5^{-/-} and B3galt5^{ΔIEC} mice, as evidenced by increased mRNA and protein levels of hepatic inflammatory cytokines (**Figure 5a, 5b, 5e and Figure 7i-j**). Furthermore, B3galt5^{-/-} and B3galt5^{ΔIEC} mice displayed more severe liver steatosis as shown by elevated liver injury and increased hepatic lipid deposition (**Figure S7a-b and Figure S10c-d**).

8. L165. What means "possibly as a mechanism in PXR dependent metabolic disorders"?

Response: Thanks for the reviewer's comments. We have changed the description on L65 as follows (Line 149-150): possibly playing an important role in obesity and as a downstream regulator of PXR.

9. Fig 3 h-i: More clinical data on the human population is required. Most of the patients were only overweight but not obese. Which ones of Table S2 were selected for Fig. 3.? Did e.g. BMI

correlate with β 3gal5 expression in colon?

Response: Thanks for the reviewer's comments. As suggested, we have added the population of healthy individuals and obese patients in **Figure 3h-i** with detail information listed in Supplementary Table 2. According to clinical guidelines²⁵, individuals in the Chinese population with a BMI of 28.0 kg/m² or higher were classified as obese. In Figure 3j, samples from patients with ID11-15 and ID27-31 were chosen for the detection of the B3gal5 protein levels. We also performed the correlation analysis of BMI and intestinal B3gal5 expression and found that BMI was negatively correlated with B3gal5 expression in the colon (**Figure 3j**).

10. L52 ... which leads to increased intestinal permeability and the transference of microbial molecules into systemic circulation.

Into the portal vein blood and liver rather than into systemic circulation. The liver is a clearance organ for most microbial molecules.

Response: Thanks for the reviewer's comments. We have corrected the description on L54 as follows: which leads to increased intestinal permeability and the transference of microbial molecules into the portal vein blood and liver.

11. L84-85: This sentence is not logical.

Response: Thanks for the reviewer's comments. We have corrected the sentence as follows (Line 66-67): Although PXR is highly expressed in the intestine, the function of intestinal PXR on metabolic diseases remains largely unknown.

Reviewer #3

In the current submission, the authors seek to characterize a functional link between the pregnane X receptor (PXR), within the intestine, the expression of B3gal5 and the regulation of the mucous layer in the context of HFD and metabolic dysfunction.

While aspects of the data presented are very interesting, this reviewer has a series of concerns that are outlined below:

1. Results - the authors present mucin/mucous layer images throughout the manuscript. The thickness of this layer must be quantified in each set of experiments to support the authors' conclusions.

Response: Thanks for the reviewer's comments. As suggested, we have added quantification of the mucus layer thickness in **Figure 6b, 6i and 7o** to support our conclusions.

2. Line 103 - "in intestine" should read "in the intestine"

Response: Thanks for the reviewer's comments. We have corrected this grammatical error.

3. Cyp3a11 is predominantly expressed in the small intestine, with very minimal colonic expression. Where in the intestine was the characterization of PXR activation and Cyp3a11 expression assessed?

Response: Thanks for the reviewer's constructive suggestions. The characterization of PXR

activation and Cyp3a11 expression was assessed in both small intestine and colon. As showed in **Figure R4**, PXR selective agonist PCN could also significantly increase the expression of Cyp3a11 in the colon.

Response Figure 4. Increased Cyp3a11 mRNA level in colon of WT mice treated with PCN. Data are mean \pm SEM, $n = 7$, $*P < 0.05$.

4. Figure 1J - given the non-specificity of the B3galt5 antibody (as depicted in Figure 3b, with a dominant “non-specific” band as interpreted by the author that is not apparent in Figure 1J), the authors should present uncropped blots with the appropriate labelling.

Response: Thanks for the reviewer’s comments. In Figure 2J, we detected the protein expression of B3galt5 in human cell line LS174T cells after treated with TBC and RIF. We did this blot by using the antibody specifically recognizing with human antigen (SAB1302633, Sigma-Alrich, St. Louis, MO, USA). The antibody in Figure 3b was custom-made to detect mouse B3galt5 (ABclonal Technology co. ltd, China). The whole blot picture was showed in **Figure R5**.

Response Figure 5. The uncropped blots of B3galt5 expression in LS174T cells treated with TBC and RIF for 48h.

5. Figure 3B - the authors need to include tissues from the B3galt5 KO mice in these blots to justify the interpretation/labelling the “non-specific” band. This also goes for the blots depicted in panels e and g. This reviewer is concerned that the authors are "picking and choosing" a band for quantification without demonstrating the specificity of the antibody. Given that assessing the expression of the B3galt5 is a key metric/outcome in this paper, these blots need to be redone to contain the appropriate controls to show antibody specificity.

Response: Thanks for the reviewer’s constructive comments. We have performed the western blots to validate the specificity of the B3galt5 antibody (custom-made to detect mouse B3galt5

by ABclonal Technology co. ltd, China) by using the colon tissue from WT and B3galt5 KO mice in **Figure 3b**. Figure 3e and 3g was also redone in the text adding the colon tissue from WT and B3galt5 KO mice, the complete blot was shown in **source data-uncropped blots**.

6. Figure 3i - the given the questionable specificity of the B3galt5 antibodies throughout the manuscript, the entire blot (i.e., uncropped) should be presented here.

Response: Thanks for the reviewer's comments. B3galt5 antibody used in this experiment was specifically reacted with human (SAB1302633, Sigma-Alrich, St. Louis, MO, USA). We have provided the complete blot in **Figure R6**:

Response Figure 6. The complete blot of B3galt5 expression in colon of healthy control and obese patients.

7. Figure 4a - this data in this panel are not analyzed correctly. These data should be reanalyzed with the appropriate 2-way ANOVA and post-hoc test. The same reanalysis is required for panel f.

Response: Thanks for the reviewer's comments. We have reanalyzed the data in Figure 4a and 4f with the appropriate 2-way ANOVA and post-hoc test in the revised manuscript.

8. Figure 5G - Raw flow cytometry plots indicating gating strategy need to be presented here to allow the reader to assess the quality of the data. Furthermore, absolute counts must accompany the cell percentages presented here. Gating strategy and complete methods for tissue dissociation and isolation must also be included in the methods section. As this section reads currently, it would allow a reader to replicate these studies.

Response: Thanks for the reviewer's constructive suggestions. As suggested by the reviewer, we have provided the raw flow cytometry plots indicating gating strategies and absolute counts in **Figure S6**. The gating strategy and complete methods for tissue dissociation and isolation were also added in **Supplementary Method-Flow cytometry** section.

9. Figure 6c - Having seen blots from leading experts in mucin biology over the course of the past 15 years, I can say that this blot is suspiciously clean. To rebut my concerns, the authors should present the entire blot. Furthermore, how can there be so little Muc2 protein in these

Bgalt5^{-/-} samples when there is clearly a reasonable signal in both the Alcian blue and Muc2 immunostaining? This reviewer is seriously concerned about the western blot data presented here.

Response: Thanks for the reviewer's comments. We have provided the complete blot below and in **Supplementary Materials**. The detailed methods used to detect the Muc2 protein was provided in **Supplementary Methods-Mucin extraction and PAS staining**. As for there are little Muc2 protein in B3galt5^{-/-} samples, there might be the following reasons: 1) Due to the restriction of protein loading capacity by SDS-PAGE gels, we only loaded 2.5% of total colonic mucin from one mouse to detect the Muc2 protein by Western blot. In this respect, the detection sensitivity of Alcian blue and Muc2 staining was higher than that of western blot; 2) The blot provided in Figure 6c was the short exposure, we also provided a long exposure picture in **Figure R7**; 3) The bottom of the blot was the non-specific band, which was confirmed by the PAS staining in Fig 6d-e; 4) We used fluorescent secondary antibody in Western blot and the blots was obtained by the LI-COR Odyssey System, which yields a cleaner background compared to the chemiluminescence detection method.

Response Figure 7. The complete blot of Muc2 expression in colon of WT and B3galt5^{-/-} mice.

10. Figure 6D and commentary lines 236-238 - pronase is usually isolated and purified from bacteria that are not gut commensals. Given that there are likely different proteases at play in the gut, the authors must show that either distinct protease known to be expressed by gut bacteria or proteases isolated from gut contents can degrade mucous isolated from WT and B3galt5^{-/-} mice. Without this, the authors cannot make the statements in lines 236-238.

Response: Thanks for the reviewer's important and intriguing comments. We have detected the effect of other two proteases, secreted protease of C1 esterase inhibitor (StcE, a well-studied *E. coli* mucinase), and O-glycoprotease (OgpA) from *A. muciniphila*, on the degradation of mucin isolated from WT and B3galt5^{-/-} mice in **Figure 6e**. Consistent with Figure 6d, StcE or OgpA had marginal effect on mucus from WT mice, but led to a significant reduction of high MW PAS-stained bands in B3galt5^{-/-} mice.

11. Figure 6i and lines 253-255 - to illustrate the PXR-dependent mechanism of TBC's thickening of the mucus layer, the authors must perform complementary experiments in PXR^{-/-} mice.

Response: Thanks for the reviewer's constructive comments. We have provided colonic Muc2

staining of PXR^{-/-} mice treated with vehicle and TBC. As shown in **Figure S6g**, TBC could not enhance mucus layer in PXR^{-/-} mice, indicating the PXR-dependent mechanism of TBC's thickening of the mucus layer.

12. Figure 7b-f-g - this data in these panels are not analyzed correctly. These data should be reanalyzed with the appropriate 2-way ANOVA and post-hoc test.

Response: Thanks for the reviewer's comments. As suggested, we have reanalyzed the data in Figure 7b-f-g with the appropriate 2-way ANOVA and post-hoc test in the revised manuscript.

13. Figure 7 J-L - why were the complementary flow cytometric analyses performed in these experiments? The authors must conduct additional experiments to present quantitative flow cytometry data assessing macrophages in these tissues.

Response: Thanks for the reviewer's comments. Per your suggestion, we have provided flow cytometry analysis of adipose tissue macrophages for B3galt5^{ΔIEC} mice in **Figure S10a-b**. Consistent with B3galt5^{-/-} mice, intestinal specific B3galt5 knockout could significantly increase M1 macrophages and decrease M2 macrophages in adipose tissues.

14. Lines 293-295 - if the authors are going to conclude that intestinal specific B3galt5 KO results in a compromised barrier, complementary permeability experiments are required (as presented in previous figures).

Response: Thanks for the reviewer's comments. As suggested, we detected the intestinal permeability of B3galt5^{ΔIEC} mice in the revised manuscript. As shown in **Figure 7m** and **7n**, the serum endotoxin and FITC-dextran levels were significantly increased in B3galt5^{ΔIEC} mice compared with their counterparts, supporting intestinal specific B3galt5 deficiency resulted in a compromised intestinal barrier.

15. Figure 8 - the authors need to include WT mice treated with both vehicle and TBC in this study to show that the reagents used in this experiment were actually functional. My first interpretation of the data presented here was that the TBC could just be non-functional in this study. Key controls would counter that conclusion.

Response: Thanks for the reviewer's constructive comments. As suggested, we have added WT mice treated both vehicle and TBC in **Figure 8a-b**. These results showed that TBC could alleviate HFD-induced body weight gain and ratio of fat pat weight to body weight in WT mice. B3galt5 knockout significantly exacerbated HFD-induced obesity; however, TBC treatment did not diminish these phenotypes in B3galt5^{-/-} mice, suggesting the beneficial effects of TBC depend on B3galt5.

16. Lines 305-308 - without including some key experiments using PXR^{-/-} mice, the authors really can't make the statement that there exists a PXR-B3galt5 axis in this system. Sure, they demonstrate to a certain degree that TBC activates the PXR, but they must perform reciprocal experiments (show that the absence of the PXR leads to the loss of B3galt5 induction in vivo).

Response: Thanks for the reviewer's constructive comments. As suggested, we have detected B3galt5 protein expression in the colon of WT and PXR^{-/-} mice treated with selective PXR agonist TBC in **Figure 2d**. TBC could significantly elevate B3galt5 expression in WT mice,

but showed no induction on B3galt5 in PXR^{-/-} mice, suggesting PXR-B3galt5 axis is existing in metabolic disease.

17. Methods - Luciferase Assay - PCN is a mouse selective agonist and shows little activity at the human PXR. Given that HEK293 cells are of human origin, how are the authors able to assess B3galt5 promoter activity using this experimental design? This reviewer is very concerned about these data.

Response: Thanks for the reviewer's comments. HEK 293 barely expressed the PXR. In this experiment, HEK293 cells were transiently transfected with the plasmids that overexpressing the mouse PXR and B3galt5 luciferase reporter. After 24 hours of transfection, the cells were treated with PXR mouse agonist PCN (10 μM) for another 24 hours before harvesting and detecting luciferase activity. Additional method details have been added in the **Supplementary Methods-Luciferase Assay** section.

18. Methods - Statistics - were the data assessed for normal distribution prior to statistical analysis with parametric tests? If so, this needs to be included. If not, non-parametric tests should be employed. Furthermore, N = 3 data sets can not be accurately assessed for normal distribution and should, by default, be analyzed using non-parametric approaches. The data presented in this submission should be reanalyzed accordingly.

Response: Thanks for the reviewer's comments. As suggested by the reviewer, we have reanalyzed the n = 3 data sets using non-parametric approaches and assessed data for normal distribution prior to statistical analysis with parametric tests. Relative statistics methods were supplied in **Method-Statistics**.

19. Figure 2 and RNA sequencing - what regions of the "intestine" were assessed? As mentioned in a previous comment, Cyp3a11 is minimally expressed in the colon. The methods section should be revised, along with all references to the "intestine" to be more clear as to what regions are being assessed throughout the manuscript.

Response: Thanks for raising this concern. We have detected the expression of PXR and its target gene Cyp3a11 in different intestinal segments of mice. As shown in **Figure R8**, the expression level of PXR was similar in various intestinal segments. Although Cyp3a11 was highly expressed in small intestine, but also expressed in the colon. In Figure 2a, ileum was used for RNA sequencing and revealed B3galt5 and Cyp3a11 were significantly induced upon PCN treatment. After detection of B3galt5 expression pattern, we found B3galt5 is specifically expressed in the colon with minimal expression in other tissues. Thereby, we also chose the colon tissue for RNA sequencing to determine whether PCN treatment can induce colonic expression of PXR target gene and B3galt5. Consistent with the ileum, B3galt5 and Cyp3a11 mRNA and protein expression were significantly upregulated in the colon upon PCN treatment (**Figure 2b**). We have added these results in **Figure S2a** and clarified the specific intestine in the **Supplementary Method-RNA sequencing**.

Response Figure 8. The intestinal expression profile of PXR and Cyp3a11. (a) Intestinal expression profile of PXR. (b) Expression profile of Cyp3a11.

20. Line 82 - “contribute” should read “contributes”.

Response: Thanks for the reviewer’s comments. We have corrected this mistake accordingly.

21. Lines 101-102 and throughout the manuscript - as a pharmacologist, this reviewer would never use the term “specific” to describe any chemical reagents. The authors should revise this to read “selective”. “Specific” can be used in the case of antibodies, but never in the case of chemical reagents since they nearly all show off-target effects (whether they’ve been reported or not).

Response: Thanks for the reviewer’s constructive comments. We have carefully corrected all these mistakes in the revised manuscript.

22. Methods for endotoxin and cytokine/chemokine analysis are missing from main body of the manuscript and supplementary methods, as are descriptions of pronase source and use in mucin degradation studies.

Response: Thanks for the reviewer’s comments. As suggested, we have supplied the methods for endotoxin and cytokine analysis in **Supplementary methods** and the description of pronase source and use in mucin degradation studies in **Supplementary Methods- Mucin extraction and PAS analysis**.

23. Methods - Animals - please include a reference to the approved animal use protocol number here to enhance post-approval assurances.

Response: Thanks for the reviewer’s comments. The approved animal use protocol number was added in the **Methods- Animals** as follows (Line 400): All animal protocols were approved by Sichuan University Animal Care and Use Committee (No: 20210222030).

24. There is no indication of the breeding scheme and whether littermate controls were used throughout the studies. This is an absolute must when performing studies on intestinal biology and metabolic disease given the microbiota’s involvement in regulating these systems. If littermate controls were not used, this is a flaw that constitutes rejection of the paper.

Response: Thanks for raising this important question. As suggested, we have provided the breeding scheme of B3galt5^{-/-} and B3galt5^{ΔIEC} mice in **Fig R9**. The littermate controls were used in all the animal experiments during our studies.

Response Figure 9. The breeding scheme of B3galt5^{-/-} and B3galt5^{ΔIEC} mice. (a) The breeding scheme of B3galt5^{-/-} mice. (b) The breeding scheme of B3galt5^{ΔIEC} mice.

References:

1. Araújo JR, Tomas J, Brenner C, et al. Impact of high-fat diet on the intestinal microbiota and small intestinal physiology before and after the onset of obesity. *Biochimie* 2017;141:97-106.
2. Kumar V, Kumar V, Mahajan N, et al. Mucin secretory action of capsaicin prevents high fat diet-induced gut barrier dysfunction in C57BL/6 mice colon. *Biomedicine & Pharmacotherapy* 2022;145.
3. Paone P, Cani PD. Mucus barrier, mucins and gut microbiota: the expected slimy partners? *Gut* 2020;69:2232-2243.
4. Taniguchi M, Okumura R, Matsuzaki T, et al. Sialylation shapes mucus architecture inhibiting bacterial invasion in the colon. *Mucosal Immunol* 2023;16:624-641.
5. Xing Y, Yan J, Niu Y. PXR: a center of transcriptional regulation in cancer. *Acta Pharm Sin B* 2020;10:197-206.
6. Teng S, Piquette-Miller M. Hepatoprotective role of PXR activation and MRP3 in cholic acid-induced cholestasis. *Br J Pharmacol* 2007;151:367-76.
7. Sui Y, Helsley RN, Park SH, et al. Intestinal pregnane X receptor links xenobiotic exposure and hypercholesterolemia. *Mol Endocrinol* 2015;29:765-76.
8. Kleinert M, Clemmensen C, Hofmann SM, et al. Animal models of obesity and diabetes mellitus. *Nat Rev Endocrinol* 2018;14:140-162.
9. Hariri N, Thibault L. High-fat diet-induced obesity in animal models. *Nutr Res Rev* 2010;23:270-99.
10. Preguiça I, Alves A, Nunes S, et al. Diet-induced rodent models of obesity-related metabolic disorders-A guide to a translational perspective. *Obes Rev* 2020;21:e13081.
11. Karpale M, Kumm O, Kärkkäinen O, et al. Pregnane X receptor activation remodels glucose metabolism to promote NAFLD development in obese mice. *Mol Metab* 2023;76:101779.
12. Ghose R, Omoluabi O, Gandhi A, et al. Role of high-fat diet in regulation of gene expression of drug metabolizing enzymes and transporters. *Life Sci* 2011;89:57-64.
13. Volynets V, Louis S, Pretz D, et al. Intestinal Barrier Function and the Gut Microbiome Are Differentially Affected in Mice Fed a Western-Style Diet or Drinking Water Supplemented with Fructose. *J Nutr* 2017;147:770-780.
14. Kodama S, Koike C, Negishi M, et al. Nuclear receptors CAR and PXR cross talk with FOXO1

- to regulate genes that encode drug-metabolizing and gluconeogenic enzymes. *Mol Cell Biol* 2004;24:7931-40.
15. Gotoh S, Negishi M. Serum- and glucocorticoid-regulated kinase 2 determines drug-activated pregnane X receptor to induce gluconeogenesis in human liver cells. *J Pharmacol Exp Ther* 2014;348:131-40.
 16. Nakamura K, Moore R, Negishi M, et al. Nuclear pregnane X receptor cross-talk with FoxA2 to mediate drug-induced regulation of lipid metabolism in fasting mouse liver. *J Biol Chem* 2007;282:9768-9776.
 17. Hassani-Nezhad-Gashti F, Rysa J, Kummu O, et al. Activation of nuclear receptor PXR impairs glucose tolerance and dysregulates GLUT2 expression and subcellular localization in liver. *Biochem Pharmacol* 2018;148:253-264.
 18. Jinhan He JG, Meishu Xu, Songrong Ren, Maja Stefanovic-Racic, Robert Martin O'Doherty, Wen Xie. PXR ablation alleviates diet-induced and genetic obesity and insulin resistance in mice. *Diabetes* 2013;62(6):1876-87.
 19. Kim S, Choi S, Dutta M, et al. Pregnane X receptor exacerbates nonalcoholic fatty liver disease accompanied by obesity- and inflammation-prone gut microbiome signature. *Biochem Pharmacol* 2021;193:114698.
 20. Ma Y, Liu D. Activation of pregnane X receptor by pregnenolone 16 alpha-carbonitrile prevents high-fat diet-induced obesity in AKR/J mice. *PLoS One* 2012;7:e38734.
 21. Zhou J, Zhai Y, Mu Y, et al. A novel pregnane X receptor-mediated and sterol regulatory element-binding protein-independent lipogenic pathway. *J Biol Chem* 2006;281:15013-20.
 22. Simon GL, Gorbach SL. Intestinal flora in health and disease. *Gastroenterology* 1984;86:174-93.
 23. Grosheva I, Zheng D, Levy M, et al. High-Throughput Screen Identifies Host and Microbiota Regulators of Intestinal Barrier Function. *Gastroenterology* 2020;159:1807-1823.
 24. Jin M, Zhang H, Wu M, et al. Colonic interleukin-22 protects intestinal mucosal barrier and microbiota abundance in severe acute pancreatitis. *Faseb j* 2022;36:e22174.
 25. Zeng Q, Li N, Pan X-F, et al. Clinical management and treatment of obesity in China. *The Lancet Diabetes & Endocrinology* 2021;9:393-405.

REVIEWERS' COMMENTS

Reviewer #1 (Remarks to the Author):

The authors responded to the reviewer's comments. However, the reviewer still has concerns about the generation of KO mice. The authors performed WB to confirm the deficiency of B3galt5 protein expression. But there is no description for the information of Ab.

The Ab for Muc2 in immunohistochemistry is not described in Methods.

Reviewer #2 (Remarks to the Author):

Although the paper has definitely improved by the revision, there are still some issues that need to be addressed, e.g. The following ones and also some of reviewer #3.

Reviewer1point1: Response Fig. 1 should be included in the paper as a supplement.

Reviewer1point3: it is Ref. 58, not 59 !

Reviewer2point1: Response Figure 2 should be included in the paper as a supplement.

Reviewer2point2: the added information is helpful; however serum level of endotoxin are of less values compared to portal vein blood levels. If possible, the authors should measure LPS levels in portal vein blood, or at least discuss the limitations of LPS measurements in peripheral blood. It is surprising that the authors, despite liver clearance of LPS, still detected some differences in the KO mice.

Reviewer2point3: This discussion should be included into the manuscript.

Reviewer #3 (Remarks to the Author):

To the authors,

Thank you for addressing my concerns.

Answers to Reviewers' Comments

Reviewer #1

Reviewer #1 (Remarks to the Author):

The authors responded to the reviewer's comments. However, the reviewer still has concerns about the generation of KO mice. The authors performed WB to confirm the deficiency of B3galt5 protein expression. But there is no description for the information of Ab.

Response: Thanks for the reviewer's comments. The detail description for the B3galt5 antibody information has been provided in **Table S4**.

The Ab for Muc2 in immunohistochemistry is not described in Methods.

Response: Thanks for the reviewer's comments. The information for Muc2 antibody in immunofluorescence has been described in **Supplementary Method- Histological analysis and immunostaining** and **Table S4**.

Reviewer #2

Although the paper has definitely improved by the revision, there are still some issues that need to be addressed, e.g. The following ones and also some of reviewer #3.

Reviewer1point1: Response Fig. 1 should be included in the paper as a supplement.

Response: Thanks for the reviewer's comments. We have added Response Fig. 1 in **Fig. S7c-d**.

Reviewer1point3: it is Ref. 58, not 59!

Response: Thanks for the reviewer's comments. We have confirmed this reference and updated its number in the revised manuscript.

Reviewer2point1: Response Figure 2 should be included in the paper as a supplement.

Response: Thanks for the reviewer's comments. We have added Response Fig. 2 in **Fig. S3a-b** in the revised manuscript.

Reviewer2point2: the added information is helpful; however serum level of endotoxin are of less values compared to portal vein blood levels. If possible, the authors should measure LPS levels in portal vein blood, or at least discuss the limitations of LPS measurements in peripheral blood. It is surprising that the authors, despite liver clearance of LPS, still detected some differences in the KO mice.

Response: Thanks for the reviewer's constructive comments. The serum we have used to test for LPS levels were portal vein blood, not whole blood. We have carefully corrected the description (Line 261 and Line 305) and annotations (Figure 6g and Figure 7m) in the revised manuscript.

Reviewer2point3: This discussion should be included into the manuscript.

Response: Thanks for the reviewer's comments. We have added this discussion in the "Discussion" of the revised manuscript as follows (Line 369-375): And the down-regulation of B3galt5 could be a result of decreased expression of PXR. Previous study has showed HFD could impact the expression of PXR and its downstream target gene Cyp3a11¹. High fat diet feeding could significantly decrease the PXR expression in liver². We also found that HFD could also significantly decrease PXR level in colon (not show in this study). In this study, we have confirmed B3galt5 was a novel target gene of PXR and can be induced by PXR activation. During HFD, the downregulated PXR might decrease B3galt5 expression, which might be assumed in humans.

References:

1. Karpale M, Kummu O, Kärkkäinen O, et al. Pregnane X receptor activation remodels glucose metabolism to promote NAFLD development in obese mice. *Mol Metab* 2023;76:101779.
2. Ghose R, Omoluabi O, Gandhi A, et al. Role of high-fat diet in regulation of gene expression of drug metabolizing enzymes and transporters. *Life Sci* 2011;89:57-64.